# Noncanonical assembly, neddylation and chimeric cullin–RING/RBR ubiquitylation by the 1.8 MDa CUL9 E3 ligase complex

Daniel Horn-Ghetko [1,7], Linus V. M. Hopf [1,2,7], Ishita Tripathi-Giesgen[1,2], Jiale Du [1], Sebastian Kostrhon[1], D. Tung Vu[1,3], Viola Beier[1], Barbara Steigenberger[4], J. Rajan Prabu [1], Luca Stier[1,2], Elias M. Bruss [1,2], Matthias Mann [3], Yue Xiong [5,6] & Brenda A. Schulman [1,2] ✉

Ubiquitin ligation is typically executed by hallmark E3 catalytic domains. Two such domains, 'cullin–RING' and 'RBR', are individually found in several hundred human E3 ligases, and collaborate with E2 enzymes to catalyze ubiquitylation. However, the vertebrate-specific CUL9 complex with RBX1 (also called ROC1), of interest due to its tumor suppressive interaction with TP53, uniquely encompasses both cullin–RING and RBR domains. Here, cryo-EM, biochemistry and cellular assays elucidate a 1.8-MDa hexameric human CUL9–RBX1 assembly. Within one dimeric subcomplex, an E2-bound RBR domain is activated by neddylation of its own cullin domain and positioning from the adjacent CUL9–RBX1 in *trans*. Our data show CUL9 as unique among RBX1-bound cullins in dependence on the metazoan-specific UBE2F neddylation enzyme, while the RBR domain protects it from deneddylation. Substrates are recruited to various upstream domains, while ubiquitylation relies on both CUL9's neddylated cullin and RBR domains achieving self-assembled and chimeric cullin–RING/RBR E3 ligase activity.

Ubiquitin is typically ligated to substrates by E1–E2–E3 tri-enzyme cascades. Recently, variant ubiquitylation cascades have been elucidated. 'E2–E3-hybrid' enzymes encompass an E2 domain that transfers ubiquitin, and other domains recruiting substrates and regulators[1–4]. Some ubiquitylation pathways involve not one, but two distinct E3 enzymes acting in series[5] or in a singular complex[6–10]. However, the vertebrate-specific CUL9 is unique in encompassing two distinct types of E3 ligase within the same polypeptide.

CUL9 (also known as PARC or H7-AP1) was originally identified as a cytoplasmic TP53-binding protein[11]. CUL9 regulates DNA damage responses, cell proliferation and apoptosis, and is a haploinsufficient tumor suppressor acting through TP53 (refs. 11–17). CUL9's CPH domain binds TP53 (refs. 18,19). This interaction determines the known CUL9-dependent cellular phenotypes[17]. CUL9 monoubiquitylates TP53 (refs. 11), but does not trigger TP53 degradation[11–17]. Yet how CUL9 achieves E3 ligase activity remains unclear. CUL9 was named based on sequence similarity to canonical cullin proteins, which serve as adapters within multiprotein cullin–RING ligases (CRLs). One end of a cullin binds an RBX-family RING protein (RBX1 for CUL1, CUL2, CUL3 and CUL4A/B, and RBX2 for CUL5). At the other end, a canonical cullin's N-terminal 'CR1' (Cullin Repeat 1) domain binds a receptor that recruits substrates for ubiquitylation. Although CUL9 binds RBX1, it lacks a CR1 domain. Thus, CUL9–RBX1 cannot regulate substrates as a canonical CRL.

[1]Department of Molecular Machines and Signaling, Max Planck Institute of Biochemistry, Martinsried, Germany. [2]Department of Chemistry, TUM School of Natural Sciences, Garching, Germany. [3]Department of Proteomics and Signal Transduction, Max Planck Institute of Biochemistry, Martinsried, Germany. [4]Mass Spectrometry Core Facility, Max Planck Institute of Biochemistry, Martinsried, Germany. [5]Department of Biochemistry and Biophysics, Lineberger Comprehensive Cancer Center, University of North Carolina at Chapel Hill, Chapel Hill, NC, USA. [6]Present address: Cullgen Inc., San Diego, CA, USA. [7]These authors contributed equally: Daniel Horn-Ghetko, Linus V. M. Hopf. ✉e-mail: schulman@biochem.mpg.de

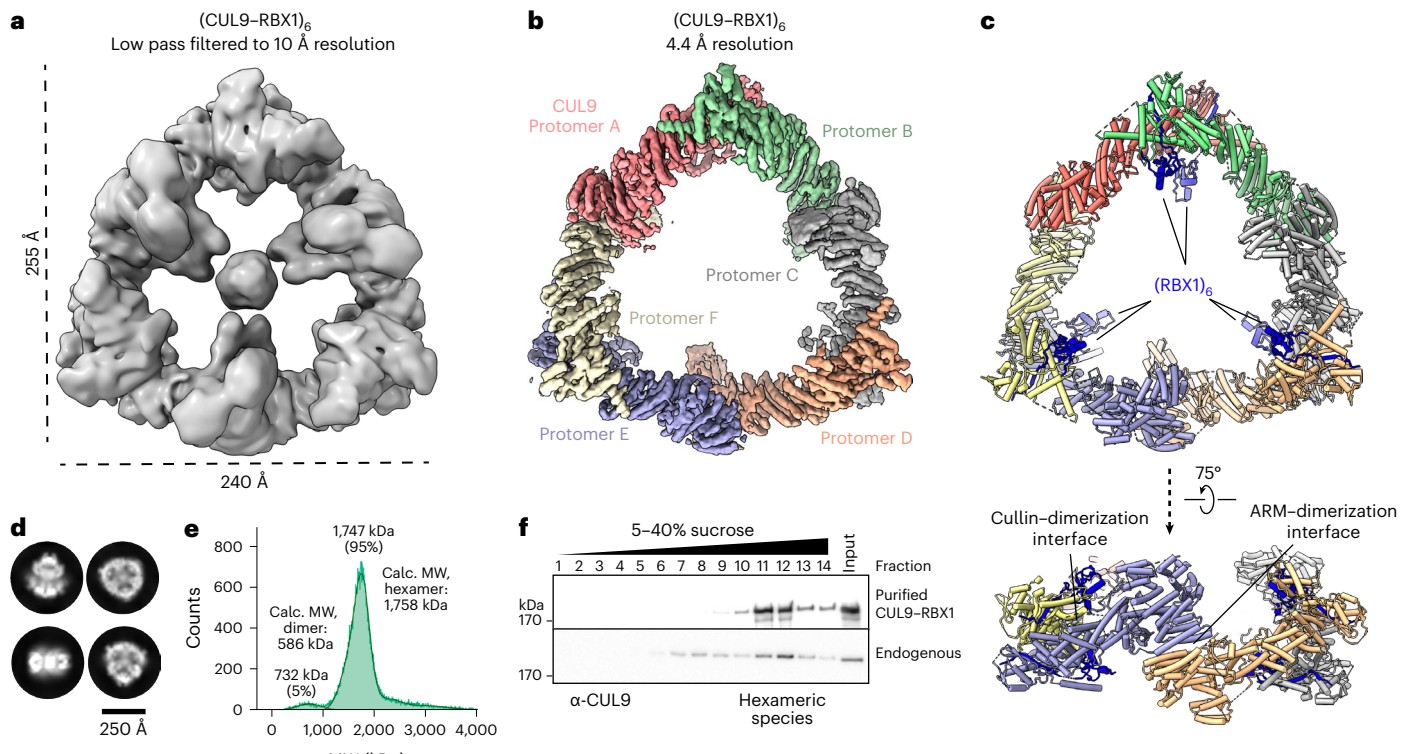

**Fig. 1 | Cryo-EM structure reveals hexameric CUL9–RBX1 E3 ligase complex.**
**a**, Cryo-EM map of CUL9–RBX1 after 3D refinement, calculated with C3 symmetry and low-pass filtered to 10 Å resolution. **b**, Cryo-EM reconstruction of hexameric CUL9–RBX1 refined to 4.4 Å resolution (calculated with C3 symmetry and postprocessed with DeepEMhancer). Individual CUL9 protomers are color-coded. **c**, Structure of hexameric CUL9–RBX1 aligned with orientation of cryo-EM map as in **b**. **d**, Representative 2D classes of CUL9–RBX1. Scale bar, 250 Å. **e**, Mass photometry analysis of purified CUL9–RBX1. Calc., calculated; MW, molecular weight. **f**, Immunoblot of CUL9 from sucrose gradient fractions of purified hexameric CUL9–RBX1 or endogenous CUL9 from U2OS cells ($n$ = 3 technically independent experiments).

*CUL9* evolved through fusion between gene duplications of the other vertebrate-specific cullin, *CUL7*, and an *ARIH*-family gene. CUL9's N-terminal 1978 residues are 63% identical to CUL7, with armadillo (ARM)- motifs, the TP53-binding CPH domain, a DOC domain and the cullin region[18–20]. Like CUL9, CUL7 lacks a CR1 domain, but instead can attain E3 ligase activity through noncanonical CRL–CRL partnership[21]. Rather than CUL7–RBX1 serving as the catalytic module like other cullin–RING complexes, it binds the TP53 substrate in a CRL7^FBXW8 complex: CUL7 binds FBXW8, exposing SKP1 to recruit NEDD8-modified CUL1–RBX1, which promotes ubiquitylation of CUL7-bound TP53 (refs. 21). CUL7 also binds CUL9. However, CUL9 does not bind FBXW8 or connect to a neddylated CUL1-based E3 ligase, and there are no clues for how it could bind CUL7.

CUL9 is unique among cullins in having a C-terminal ARIH-family RBR E3 ligase domain[20,22]. RBR E3s catalyze ubiquitylation in E1–E2–E3 cascades[23–25]. After receiving ubiquitin from E1, a ubiquitin-loaded E2 binds the RBR E3's RING1 domain. Ubiquitin is transferred from the RING1-bound E2 to the catalytic cysteine in the RBR E3's Rcat domain, which then transfers ubiquitin to substrates. Evolutionary precursors of CUL9's RBR element are ARIH1 and ARIH2, which on their own are autoinhibited by their distinctive 'Ariadne' domain sequestering the Rcat domain[10,26]. These ARIH-family RBR E3s become active when their Ariadne domains bind a cognate RBX RING domain and neddylated canonical cullin (ARIH1 with RBX1 and neddylated CUL1, CUL2 or CUL3, and ARIH2 with RBX2 and neddylated CUL5)[7,9,10,27]. CUL9's combination of cullin and ARIH-family RBR domains hint at a similar CRL–RBR ubiquitylation mechanism, albeit without a CRL substrate receptor.

The canonical CRL–ARIH E3–E3 mechanism relies on cullin neddylation, a process akin to ubiquitylation, with distinct E2s attaching NEDD8

to cullins[28–30]. It is currently thought that cellular neddylation involves the early-evolving E2, UBE2M, modifying RBX1-bound cullins (that is, CULs1–4), while the late-evolving E2, UBE2F, modifies RBX2-bound CUL5 (refs. 31–34). UBE2F can neddylate all these cullin–RING complexes in vitro[31]. In the absence of substrate, NEDD8 is removed from conventional cullins by the COP9 signalosome (CSN)[29,30,35–38]. Despite progress in understanding neddylation pathways and how they are regulated for conventional CRLs, the CUL9 neddylation pathway and functional consequences of CUL9 neddylation remain unknown.

With 2,517 residues, CUL9 is the largest cullin and RBR protein, and among the dozen largest E3 ligase subunits, and its structural mechanisms have remained elusive. Here, cryo-EM, cellular studies and biochemistry reveal CUL9–RBX1 forms a unique, 1.8-MDa oligomeric assembly, with a distinct neddylation pathway and chimeric CRL–RBR E3 ubiquitin ligase activity.

## Results

### CUL9–RBX1 forms a 1.8-MDa hexameric triangular assembly

Size-exclusion chromatography of recombinant CUL9–RBX1 (expressed in human embryonic kidney 293S (HEK293S) cells) suggested a much larger complex than its calculated molecular weight of 293 kDa (Extended Data Fig. 1a). Accordingly, a cryo-EM map revealed its triangular-shaped assembly with 240 Å-long vertices, measuring 255 Å across, with six inward-facing globular domains (Fig. 1a). Three-dimensional (3D)-variability analysis showed heterogeneity of the central domains, explaining their low resolution and flexibility of the outer triangular scaffold (Supplementary Video 1).

Refinement, applying C3 symmetry, yielded a 4.4 Å resolution map revealing secondary structures for the triangular scaffold. The map

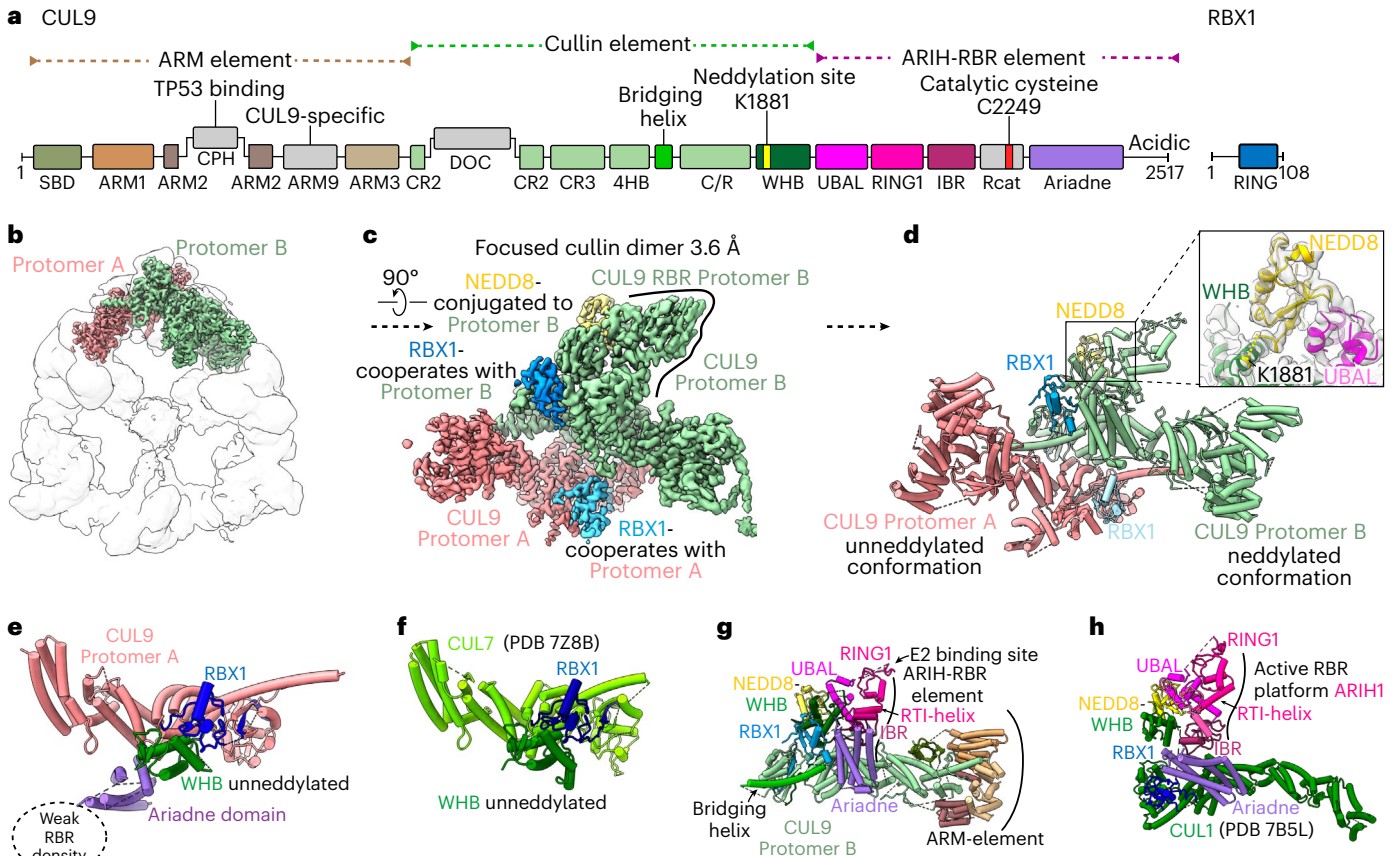

**Fig. 2 | Distinct architectures for unneddylated and neddylated CUL9–RBX1 protomers. a**, Domains of CUL9 and RBX1. Domains in gray were not directly assigned in cryo-EM density. **b**, Cryo-EM map focused on the cullin dimer between CUL9–RBX1 protomers A and B. Individual protomers are color-coded and displayed within the global map over the hexamer. **c**, Close-up of focused cullin dimer cryo-EM map refined to 3.6 Å (postprocessed with DeepEMhancer). CUL9 protomers, NEDD8 and two RBX1 units colored individually. RBR domain of protomer B is highlighted with an adjacent line. **d**, Structure of dimeric CUL9–RBX1 assembly, displaying unneddylated (protomer A) and neddylated (protomer B) conformations. The close-up shows NEDD8 linked to CUL9's

WHB domain, and the NEDD8-interacting CUL9 RBR element UBAL domain in cryo-EM density. **e**, Part of the unneddylated CUL9–RBX1 protomer A. **f**, Part of CUL7–RBX1 from CRL7$^{FBXW8}$ (PDB 7Z8B, SKP1-FBXW8 hidden) corresponding to the region of CUL–RBX1 shown in **e**. CUL9 and CUL7 structures in **e** and **f** are aligned on the WHB domain. Trajectory of weak RBR domain density for CUL9–RBX1 protomer A indicated by the circle. **g**, Structure of neddylated CUL9–RBX1 protomer B with domains colored as in **a**. **h**, Structure of neddylated CRL1-ARIH1 E3–E3 super assembly (PDB 7B5L, UBE2L3, ubiquitin, SKP1, SKP2, CKSHS1, p27, cyclin A and CDK2 hidden). CR2 and CR3 domains of CUL1 are aligned on the corresponding domains of CUL9 in **g**.

readily fit AlphaFold2 (ref. 39)-predicted models for elements from six CUL9–RBX1 protomers (Fig. 1a–d and Extended Data Fig. 1b). Mass photometry and size-exclusion chromatography–multi-angle light scattering (SEC–MALS) validated the CUL9–RBX1 hexamer (1.8 MDa, Fig. 1e and Extended Data Fig. 1c). We examined whether CUL9 forms such an oligomer in cells with sucrose gradient fractionation of U2OS cell lysates. Endogenous CUL9 migrates in the same fractions as purified hexameric CUL9, with some in preceding fractions consistent with the dimer also detected by mass photometry (Fig. 1e,f and Extended Data Fig. 1d).

While some CRL substrate receptors oligomerize (for examples, see refs. 40–45), CUL9–RBX1 is unique among cullin–RING complexes in self-mediating assembly. Both ends of each protomer connect to another (Fig. 2a). One interface involves the cullin element from two protomers. This cullin dimer adopts a boomerang shape. The bends in the three boomerangs are the corners in the triangular hexamer (Fig. 1b,c). The second dimerization interface occurs in the center of each side of the triangle where ARM1 domains of two protomers interact (Fig. 2b–d and Extended Data Fig. 1e). The following sections describe assigning positions of elements beyond the triangular scaffold, and high-resolution visualization of the unique ubiquitin ligase elements within the hexameric CUL9–RBX1 assembly.

## Distinct neddylated and unneddylated CUL9–RBX1 conformations

Focused classification yielded high-resolution insights into the domains (Fig. 2a). A subset of roughly 70,000 particles displayed additional density adjacent to the cullin dimerization interface. A 3.6 Å resolution map over this region allowed building and refining atomic models for the two CUL9–RBX1 protomers in a dimeric subcomplex (Fig. 2b–d, Extended Data Fig. 2 and Table 1). The visible regions from CUL9 include the small beta domain (SBD), the ARM element (ARM1–ARM3 domains), the cullin element (CR2, CR3, 4HB, C/R and WHB domains) and ARIH1-RBR element. RBX1 has two domains: an N-terminal strand embedded in CUL9's C/R domain is tethered to the C-terminal RING domain. Fitting the dimeric subcomplex into the full map showed details of the hexameric assembly (Figs. 1c and 2a–d).

Although the two CUL9–RBX1 protomers in this refined map, referred to as A and B, superimpose over most of their length, they diverge in arrangement of some CRL (CUL9 WHB and RBX1 RING domains) and all ARIH-RBR elements (Fig. 2b and Extended Data Fig. 3a). Notably, corresponding regions in their canonical counterparts rearrange during ubiquitylation reactions[9,10]. CUL9's RBR E3 catalytic cysteine-containing Rcat domain was not visible for either

**Table 1 | Cryo-EM data collection, refinement and validation statistics**

| | CUL9–RBX1 focused cullin dimer (neddylated + unneddylated) | CUL9–RBX1 hexameric assembly | CUL9–RBX1 focused dimeric core | CUL9–RBX1 focused on E2-like density | CUL9–RBX1 symmetry expanded unneddylated dimer |
|---|---|---|---|---|---|
| CUL9 | WT | WT | WT | WT | WT |
| RBX1 | Residues: 5–108 | Residues: 5–108 | Residues: 5–108 | Residues: 5–108 | Residues: 5–108 |
| NEDD8 | Endogenous | | | | |
| E2 | Endogenous | | | | |
| **Accession codes** | EMD-18216, PDB 8Q7H | EMD-18214, PDB 8Q7E | EMD-18218 | EMD-18217 | EMD-19179, PDB 8RHZ |
| **Data collection and processing** | | | | | |
| Microscope, magnification | Krios, 105,000 | Krios, 105,000 | Krios, 105,000 | Krios, 105,000 | Krios, 105,000 |
| Voltage (kV) | 300 | 300 | 300 | 300 | 300 |
| Electron exposure (e⁻/Å²) | ~60 | ~60 | ~60 | ~60 | ~60 |
| Defocus range (μm) | ~−0.7 to −2.8 | ~−0.7 to −2.8 | ~−0.7 to −2.8 | ~−0.7 to −2.8 | ~−0.7 to −2.8 |
| Pixel size (Å) | 0.8512 | 0.8512 | 0.8512 | 0.8512 | 0.8512 |
| Symmetry imposed | C1 | C3 | C1 | C1 | C3 |
| Initial particle images (no.) | 1,212,742 | | | | |
| Final particle images (no.) | 71,928 | 611,252 | 71,928 | 153,970 | 661,706 |
| Map resolution (Å) | 4.1 | 4.4 | 3.5 | 4.2 | 3.37 |
| FSC threshold | (0.143) | (0.143) | (0.143) | (0.143) | (0.143) |
| Map resolution range (Å) | 3.5–9 | 3.6–13 | | | 2.8–15 |
| **Refinement** | | | | | |
| Initial model used (PDB code) | Alphafold Q8IWT37 PDB 7B5L PDB 7Z8B | 8Q7H | | | 8Q7H |
| Model resolution (Å) | 4.1 | 4.4 | | | 3.37 |
| FSC threshold | (0.143) | (0.143) | | | (0.143) |
| Model resolution range (Å) | 3.5–9 | 3.6–13 | | | 2.8–15 |
| Map sharpening *B* factor (Å²) | −127 | −150 | | | −139.7 |
| Model composition | | | | | |
| Nonhydrogen atoms | 18,033 | 29,815 | | | 15,223 |
| Protein residues | 2,932 | 6,016 | | | 2,439 |
| Ligands | 10 | | | | 6 |
| *B* factors (Å²) | | | | | |
| Protein | 104.9 | 90.8 | | | 98.28 |
| Ligand | 296.65 | | | | 252.68 |
| R.m.s. deviations | | | | | |
| Bond lengths (Å) | 0.005 | 0.005 | | | 0.008 |
| Bond angles (°) | 1.009 | 1.071 | | | 1.033 |
| **Validation** | | | | | |
| MolProbity score | 1.34 | 0.84 | | | 1.67 |
| Clashscore | 6.13 | 1.23 | | | 4.73 |
| Poor rotamers (%) | 0 | 0 | | | 0.12 |
| Ramachandran plot | | | | | |
| Favored (%) | 99.21 | 99.4 | | | 93.47 |
| Allowed (%) | 0.79 | 0.6 | | | 6.53 |
| Disallowed (%) | 0 | 0 | | | 0 |

protomer. However, the remainder of ARIH-RBR domains were visible in protomer B, while only the Ariadne domain—a key regulatory element—was observed in protomer A.

The structure showed the cullin dimerization interface in detail. The cullin CR2 domain and subsequent regions pack in a head-to-tail orientation as a pseudosymmetric 60 Å long unit. At the center, the two CUL9 4HB domains interact. A 40-residue long helix, which we term 'bridging helix', radiates outward from each 4HB, bridges the subsequent heterodimeric cullin/RBX (C/R) domain, and culminates by packing against the CR2 domain from the opposite protomer

(Extended Data Fig. 3b). Additionally, protomer B's Ariadne domain approaches RBX1 from the opposite protomer (A), but not vice versa.

The disparate conformations of the two protomers arises from a striking difference in their molecular composition: protomer A is unneddylated; CUL9's K1881 is modified by NEDD8 in protomer B. CUL9 K1881 corresponds to the WHB domain site that is neddylated in canonical cullins. NEDD8's appearance was fortuitous, because we did not enzymatically neddylate CUL9–RBX1 in vitro before cryo-EM analysis. Superimposing homologous regions of protomer A (unneddylated) on protomer B (neddylated) and vice versa show that the hexamer could be formed by either a fully unneddylated or neddylated complex, the former also observed in a map obtained through symmetry expansion (Extended Data Fig. 3c,d).

The unneddylated protomer A represents an inactive form of CUL9–RBX1, resembling the previous structure of CUL7–RBX1 (ref. 21) (Fig. 2e,f). Moreover, CUL9's WHB domain has high sequence similarity to CUL7 (Extended Data Fig. 3e), which so far has not been shown to be neddylated[12,21]. The arrangement of cullin elements and RBX1 in the previous CUL7–RBX1 structure, and CUL9–RBX1 protomer A, blocks the RING domain and thus appears incompatible with either neddylation or ubiquitylation[21,46] (Extended Data Fig. 4a).

The neddylated protomer B shows an activated conformation. NEDD8 and its covalently linked CUL9 WHB domain wedge between RBX1's RING and CUL9's RBR domain. NEDD8's I44 patch binds CUL9's UBA-like (UBAL) domain to mold the RBR domain into the active E3 configuration[9,10,25]. This includes the emblematic straight conformation for the RING1-to-IBR (RTI) helix, which contributes to active RBR E3 platforms (Fig. 2g, h)[25,47,48].

Comparing the two protomers showed how neddylation transforms the conformation of CUL9–RBX1. First, the neddylated WHB domain is rotated roughly 35° and translated about 10 Å away from the cullin scaffold (Extended Data Fig. 4b). This WHB domain repositioning avoids clashing with the IBR and Ariadne domains in the ARIH-RBR element, and promotes positioning of the IBR domain by the activated semicircular layered arrangement of CUL9's Ariadne domain, RBX1's RING domain, CUL9's WHB domain, NEDD8 and CUL9's UBAL domain (Extended Data Fig. 4c). Second, RBX1's RING domain has rotated around 160° relative to the C/R domain. Instead of RBX1's C terminus tucking into a WHB domain groove in unneddylated CUL9 (as also observed for CUL7, ref. 21), it packs against a CUL9 C/R domain loop visible only in the neddylated protomer (Extended Data Fig. 4d). To our knowledge, RBX1's extreme C terminus has not been visualized in a canonical CRL, but it makes distinct interactions with CUL9 depending on neddylation status. Finally, in the neddylated protomer B, the RBX1 RING anchors the Ariadne domain (Fig. 2c). These interactions resemble RBX1 and RBX2 RING domains binding to activated ARIH1 and ARIH2 Ariadne domains, respectively[9,10]. However, CUL9's Ariadne domain helices are shorter and relatively twisted, and uniquely interact with CUL9's cullin element adjacent to the dimerization interface (Extended Data Fig. 4e).

## EM density shows an E2 bound to neddylated CUL9's RBR RING1

The neddylated protomer B showed additional density associated with the RING1 domain that could not be attributed to CUL9 or RBX1 (Extended Data Fig. 2). This clearly fit an E2 ubiquitin conjugating enzyme (Fig. 3a). The density would be consistent with both families of E2 (UBE2D or UBE2L3) shown to bind RING1 domains of other RBR E3s[48,49]. We modeled the E2 as UBE2L3 based on: (1) our isothermal titration calorimetry showing UBE2L3, but not a UBE2D-family E2, binding the CUL9 RBR element; (2) affinity purification–mass spectrometry (AP–MS) data showing RING1-dependent endogenous UBE2L3 association with CUL9 ectopically expressed in HEK293S cells; (3) previous data showing CUL9 binds UBE2L3 (refs. 11,50,51) and (4) a predilection for RBR E3s to use this E2 (ref. 49) (Extended Data Fig. 5a,b).

The structural model shows UBE2L3's F63 side-chain engaging a hydrophobic surface in the CUL9 RING1 domain as in other E2-RBR E3 complexes[9,48] (Extended Data Fig. 5c). Furthermore, NEDD8, its linked CUL9 WHB domain, the CUL9 RBR element and the E2 superimpose with previous structures representing an RBR E3 reaction (ubiquitin transfer from E2 to E3) for super-assemblies between canonical CRLs and ARIH1 (ref. 9) (Fig. 2g,h). In addition to such canonical interactions with the RBR domain of protomer B, the backside of the E2 approaches the ARM3 domain of Protomer C in hexameric CUL9–RBX1, consistent with cross-linking mass spectrometry (XL-MS) for a CUL9–RBX1 complex with a stabilized mimic of the UBE2L3-ubiquitin conjugate (Extended Data Fig. 5d,e and Supplementary Table 1).

## Ubiquitylation depends on neddylated CRL and RBR features

The well-studied ARIH-family RBR E3s, ARIH1 and ARIH2, are autoinhibited by the Ariadne domain restraining the catalytic Rcat domain[10,26]. These E3s become active when their Ariadne domains bind cognate neddylated CRL E3s, which eliminates autoinhibitory intramolecular interactions[7,9,10,27,52]. Many structural features of activated E3–E3 complexes between ARIH1 or ARIH2 and canonical neddylated CRLs are observed in CUL9–RBX1. First, the lack of density corresponding to CUL9's Rcat domain suggests it is not restrained. Second, CUL9's Ariadne domain is engaged by the cullin–RING element, albeit in distinct arrangements in different protomers. Third, for the neddylated protomer, CUL9's Ariadne domain binds RBX1's RING as in active E3–E3 complexes[9,10]. Fourth, the RBR element associated with the neddylated CUL9 WHB domain superimposes with the corresponding region of ARIH1 and neddylated CUL1 (ref. 9). Accordingly, our purified CUL9–RBX1 displayed autoubiquitylation activity in vitro. Autoubiquitylation was observed with E2s in the promiscuous UBE2D-family that function with diverse E3s, and with UBE2L3 that is specialized to transfer ubiquitin to RBR E3s (ref. 49) and stably binds CUL9 (Fig. 3b and Extended Data Fig. 6a). Although the primary autoubiquitylation site resides in a loop that was not visible in the EM maps, this could in principle localize to a ubiquitin-linked Rcat domain based on their connections to the structured regions (Supplementary Table 2).

We sought to assay roles of neddylated CRL and RBR elements in ubiquitylation of a protein recruited to CUL9–RBX1. CUL9's best-recognized interaction partner is TP53 (ref. 53), which binds the CPH domain[18,19]. We observed robust CUL9–RBX1-dependent ubiquitylation of TP53 (Fig. 3b). Use of the E2 UBE2L3 in reactions led to preferential TP53 modification versus autoubiquitylation. This is reminiscent of the redirection of ARIH1 and ARIH2 activity from automodification to a neddylated CRL's receptor-bound substrate in E3–E3 super-assemblies[7,10]. Thus, the data suggested TP53 ubiquitylation could proceed through an E3–E3-like mechanism, here encompassed within neddylated CUL9–RBX1. To test this, we assayed effects of mutations eliminating key elements. The mechanistic roles of CUL9's RBR element were verified by the findings that TP53 ubiquitylation was nearly abrogated on mutation of the RBR catalytic cysteine (C2249A), or deleting the TP53-binding CPH domain, the ARIH-RBR element, the RBR RING1 domain that binds the E2, or mutation of the RING1-binding residue in UBE2L3 (F63A) (Fig. 3b and Extended Data Fig. 6b,c). Similarly, TP53 ubiquitylation was severely impaired on eliminating the neddylation site (K1881R), confirming the E3–E3-like mechanism. On the other hand, TP53 was still substantially ubiquitylated by CUL9–RBX1 deletion mutants lacking the ARM3, ARM9 or DOC domains (Fig. 3b and Extended Data Fig. 6b,c).

We considered that CUL9–RBX1 could exert regulation by mono-ubiquitylation rather than poly-ubiquitylation because CUL9 has not been found to control TP53 degradation. Furthermore, studies of CUL9's evolutionary precursor showed ARIH1 preferentially monoubiquitylates substrates recruited to neddylated CRLs[7]. To determine whether this property is shared by CUL9–RBX1, we tested TP53 modification with a fluorescently tagged, lysineless version of ubiquitin,

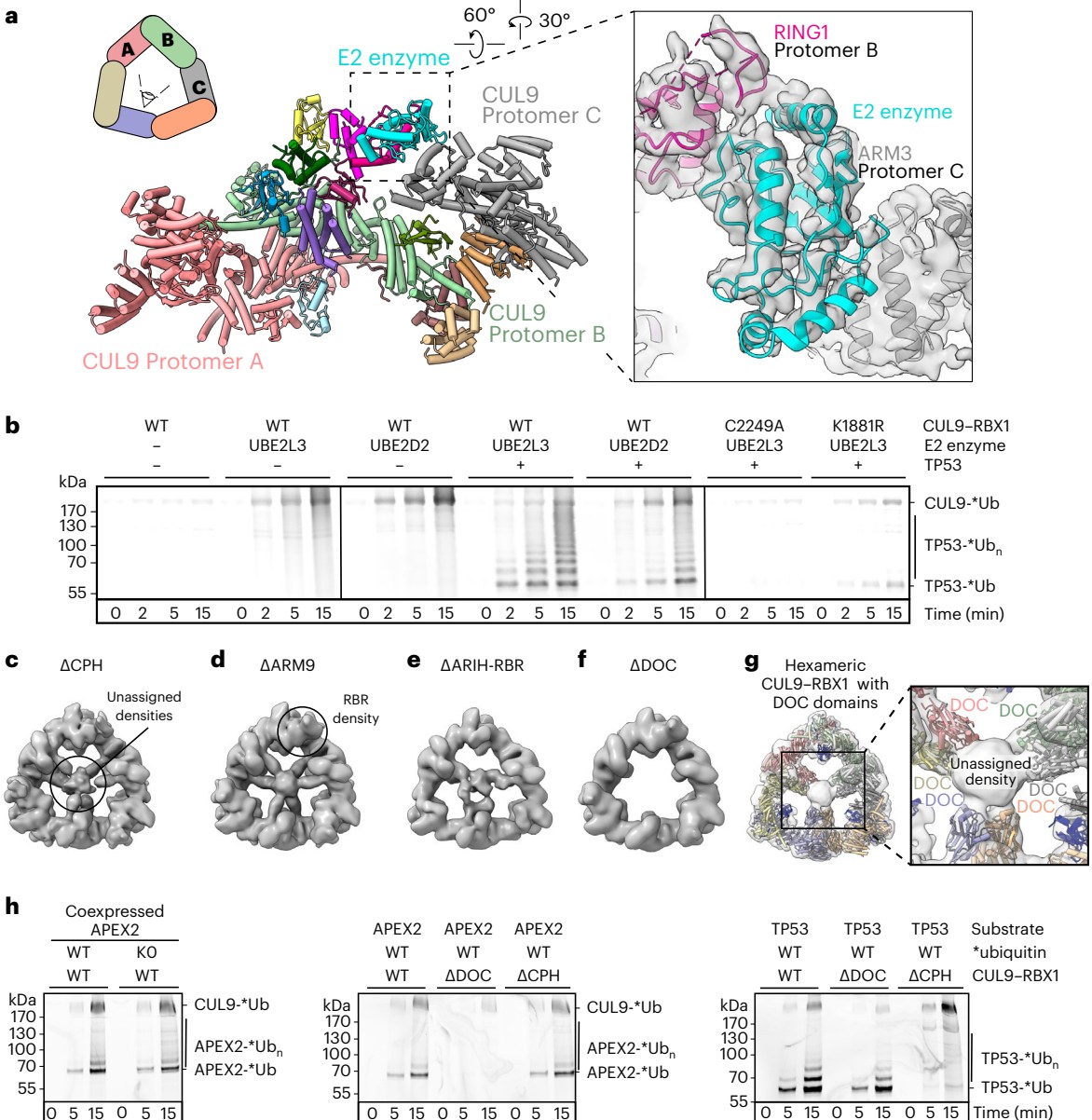

**Fig. 3 | Cryo-EM and biochemical analysis of CUL9 variants reveal E2 binding, locations of DOC domains and cullin–RING and RBR elements essential for ubiquitylation activity. a**, The top left shows a cartoon schematic of hexameric CUL9–RBX1 assembly with color-coded protomers. The center shows the structure of CUL9–RBX1 protomers A, B and C with E2 enzyme docked and colored as in Fig. 2. The right shows a close-up of RING1-E2-ARM3 interactions, displayed in cryo-EM density. Quality of density allows fitting of E2 enzyme structure but was not sufficient to determine E2 identity. **b**, In vitro ubiquitylation assays testing autoubiquitylation and activity toward substrate TP53, comparing reactions with UBE2L3 and UBE2D2 as E2s, role of CUL9 RBR Rcat with catalytic C2294A substitution and of CUL9 WHB domain neddylation with K1881R substitution. Assays detect fluorescently labeled ubiquitin (*Ub) ($n = 2$ technically independent experiments). **c**, Cryo-EM map of CUL9–RBX1 variant in which CPH domain was replaced by a GSGSGSGS linker sequence

($\Delta$CPH). For reference, unassigned central density found in WT CUL–RBX1 and in this and several other variants is circled. **d**, Cryo-EM map of CUL9–RBX1 variant in which ARM9 domain was replaced by a GSGSGSGS linker sequence ($\Delta$ARM9). For reference, density corresponding to RBR domain in WT CUL9–RBX1 and variants is circled. **e**, Cryo-EM map of CUL9–RBX1 variant lacking ARIH-RBR element ($\Delta$ARIH-RBR) by truncation at residue 1978. **f**, Cryo-EM map of CUL9$^{\Delta DOC}$–RBX1. **g**, DOC domains fitted into the unassigned central density in CUL9–RBX1 hexamer map at low threshold. **h**, In vitro ubiquitylation assays testing activity of recombinant CUL9–RBX1 and CUL9–RBX1 variants. The assays were performed with either APEX2 or TP53 as substrates, and detect fluorescently labeled ubiquitin ($n = 2$ technically independent experiments). APEX2 was either coexpressed and copurified with CUL9–RBX1, or purified and separately added as indicated.

whose extended N terminus prevents formation of linear chains (K0 *Ub) (Extended Data Fig. 6b). The similar banding pattern of reaction products on SDS–PAGE for wild-type (WT) and lysineless ubiquitin is consistent with CUL9–RBX1 mediating multi-mono-ubiquitylation of TP53, and our finding that multiple sites are modified (Supplementary Table 3).

## Roles of the CUL9 DOC domain

To assign the density in the center of the triangular scaffold, we obtained cryo-EM data for deletion mutant versions of CUL9, focusing on domains that were not visible in other maps: the CPH, ARM9, DOC and Rcat domains. The CPH domain emanates from within the ARM2 domain by roughly 40-residue connections. The ARM9

## Table 2 | Cryo-EM data collection, refinement and validation statistics

| CUL9 RBX1 | CUL9$^{\Delta CPH}$–RBX1 ΔCPH Residues: 5–108 | CUL9$^{\Delta ARM9}$–RBX1 ΔARM9 Residues: 5–108 | CUL9$^{\Delta ARIH-RBR}$–RBX1 ΔARIH-RBR Residues: 5–108 | CUL9$^{\Delta DOC}$–RBX1 ΔDOC Residues: 5–108 |
|---|---|---|---|---|
| **Accession codes** | EMD-18220 | EMD-18222 | EMD-18223 | EMD-18221 |
| **Data collection and processing** | | | | |
| Microscope | Glacios | Arctica | Arctica | Arctica |
| Magnification | 22,000 | 73,000 | 73,000 | 73,000 |
| Voltage (kV) | 200 | 200 | 200 | 200 |
| Electron exposure (e⁻/Å²) | ~60 | ~60 | ~60 | ~60 |
| Defocus range (μm) | ~−1.2 to −3.3 | ~−1.2 to −3.3 | ~−1.2 to −3.3 | ~−1.2 to −3.3 |
| Pixel size (Å) | 1.885 | 1.997 | 1.997 | 1.997 |
| Symmetry imposed | C1 | C1 | C1 | C1 |
| Initial particle images (no.) | 206,542 | 268,429 | 493,491 | 414,098 |
| Final particle images (no.) | 30,949 | 21,505 | 32,978 | 29,312 |
| Map resolution (Å) | 9.5 | 12.5 | 13.9 | 13.7 |
| FSC threshold | (0.143) | (0.143) | (0.143) | (0.143) |

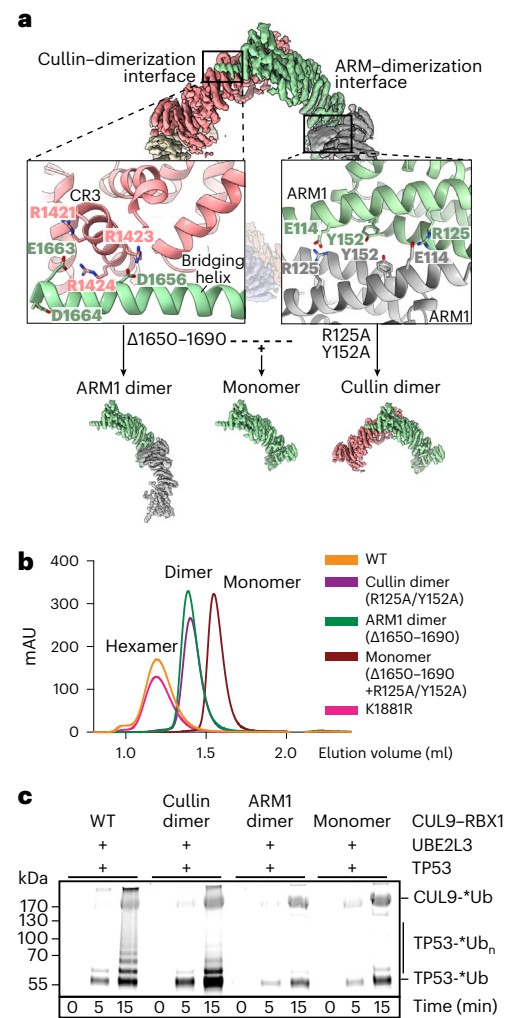

**Fig. 4 | Disruption of the oligomerization interfaces influences CUL9–RBX1's hexameric state and ubiquitylation activity. a**, Schematic of structures highlighting interfaces mediating oligomerization and how their disruption would yield dimeric or monomeric CUL9–RBX1 species. The top shows a cryo-EM map of CUL9–RBX1 hexamer with close-ups of the ARM1 dimer interface and the bridging helix–CR3 interactions at the cullin dimer interface. Each interface is present three times in the CUL9–RBX1 hexamer. The bottom shows ARM1 dimer, monomer and cullin dimer maps dissected from EM density over CUL9–RBX1 hexamer. ARM1 dimer was made by replacing residues 1650–1690 with GSGSGSGS, cullin dimer by the two point mutations (R125A Y152A) and monomer by a combination of both. **b**, Size-exclusion chromatography analysis of recombinant CUL9–RBX1 and indicated CUL9–RBX1 variants. **c**, Ubiquitylation assays testing fluorescent ubiquitin (*Ub) transfer to TP53 by indicated CUL9–RBX1 variants (n = 2 technically independent experiments).

domain, unique to CUL9, connects to the ARM2 and ARM3 domains by roughly 70- and 20-residue linkers, respectively. The DOC domain is inserted within the CR2 domain, via short tethers. Finally, the catalytic Rcat domain is thought to be flexibly tethered in activated RBR E3s (refs. 9,10). The cryo-EM maps were not overtly affected by deleting either the CPH (CUL9$^{\Delta CPH}$) or ARM9 (CUL9$^{\Delta ARM9}$) domains (Fig. 3c,d, Table 2 and Extended Data Fig. 7a,b). Although the central density seemed less ordered for CUL9$^{\Delta ARIH-RBR}$ compared to WT, it was still visible, whereas the deleted ARIH-RBR element was not (Fig. 3e, Table 2 and Extended Data Fig. 7c). This map also demonstrated that the ARIH-RBR element is not required for CUL9–RBX1 self-assembly.

Deleting CUL9's DOC domain eliminated the inward-pointing and globular density in the center (Fig. 3f, Table 2 and Extended Data Fig. 7d). Moreover, AlphaFold2-modeled DOC domains fit into the six inward-facing densities in a manner compatible with the short tethers to the cullin element (Fig. 3g). To gain insights into a potential role for the DOC domain, we compared interactors of CUL9–RBX1 versus CUL9$^{\Delta DOC}$–RBX1. Cross-referencing our AP–MS hits (Extended Data Fig. 8a) with CUL9 interactors reported by ourselves[54] and others in BioGRID[53] revealed a single top hit: APEX2. Indeed, APEX2 was ubiquitylated in vitro by neddylated CUL9–RBX1, depending on CUL9's DOC domain, neddylation site (K1881) and RBR catalytic cysteine (C2249) (Fig. 3h and Extended Data Fig. 8b). Notably, APEX2 ubiquitylation was unaffected by deletion of CUL9's CPH domain, while TP53 was subject to ubiquitylation by the CUL9$^{\Delta DOC}$–RBX1 mutant. Although future studies will be required to determine the biological functions of APEX2 ubiquitylation by CUL9–RBX1, we note that its enzymatic activity as an apurinic–apyrimidinic endodeoxyribonuclease is in line with previous findings that CUL9 plays roles in maintaining genome integrity[13,15,17].

### Oligomeric assembly contributes to substrate ubiquitylation

To determine a functional role of the higher-order assembly, we identified critical residues at each of the dimerization interfaces. The 'ARM1' interface involves the N-terminal domains from two CUL9 protomers packing against each other in opposite directions. Here, a central intermolecular hydrophobic core is stabilized by Y152 from both protomers. The edges are stabilized by a salt bridge between R125 from one protomer and E114 from the other (Fig. 4a). Meanwhile, the 'cullin' dimerization interface involves D1656, E1663 and D1664 in the bridging helix from one protomer interacting with a CR3 domain basic patch on the other. Indeed, the hexameric assembly is disrupted by mutants in the ARM1 interface (R125A Y152A), or by eliminating the cullin interface through deleting part of the bridging helix (Fig. 4a,b and Extended Data Fig. 1a). We term the former structures 'cullin dimers' due to their maintaining the cullin dimerization interface, and the latter 'ARM1 dimers' due to their maintaining the ARM1 dimerization interface. Mutation of both interfaces further shifts the migration in gel filtration chromatography, consistent with formation of a monomer (Fig. 4b).

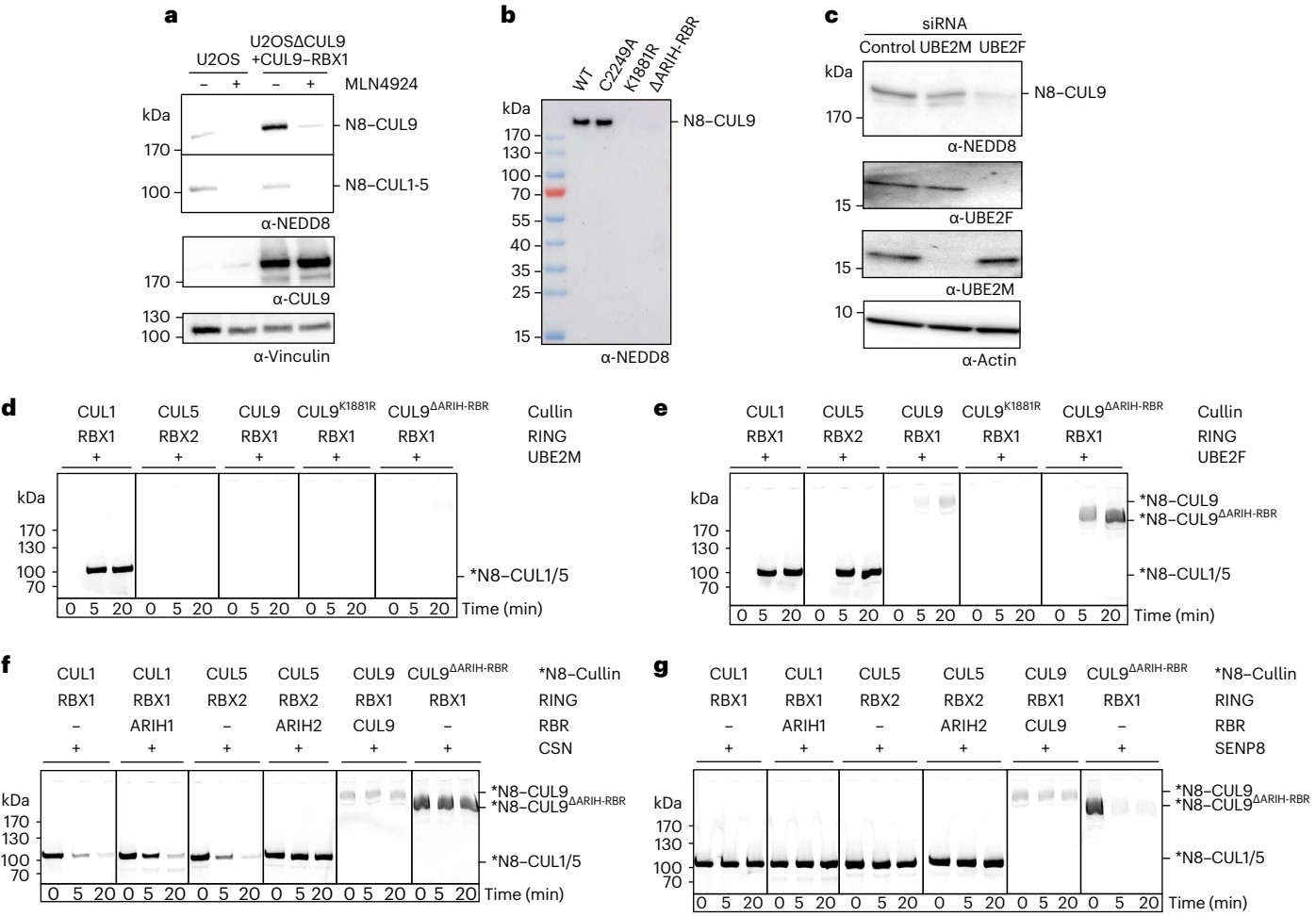

**Fig. 5 | Insights into neddylation and deneddylation of CUL9–RBX1. a**, Anti-CUL9 and anti-NEDD8 immunoblots after treatment with NAE enzyme inhibitor MLN4924 in parental U2OS cells, and CUL9 knock-out cells stably expressing CUL9 ectopically. Portions of the blot probed for neddylation correspond to CUL9 or canonical cullins (CUL1–CUL5). Immunoblot for vinculin serves as loading control. **b**, Immunoblot detecting NEDD8 shows relative modification of purified recombinant CUL9–RBX1 and indicated variants (CUL9$^{\Delta ARIH-RBR}$–RBX1 is truncated at residue 1978). **c**, Anti-NEDD8 immunoblot probing region of gel corresponding to CUL9 for U2OS cells either untreated (Control), or treated with siRNA against the neddylation E2s UBE2F or UBE2M. Other panels display immunoblots probing UBE2F, UBE2M or actin as a loading control. **d**, In vitro assays show neddylation by the E2 UBE2M, detecting fluorescent NEDD8 (*N8)

transferred to purified canonical cullin–RING complexes (CUL1–RBX1, CUL5–RBX2) as controls alongside purified CUL9–RBX1, or the K1881R variant with Arg replacement for the neddylation site, or the variant retaining the neddylation site but lacking ARIH-RBR element (ΔARIH-RBR) by truncation at residue 1978. **e**, Assays as in **d**, except with neddylation E2 UBE2F. **f**, In vitro assay probing deneddylation by CSN of the indicated fluorescently neddylated cullin–RING complexes. Effects of CUL1–RBX1 and CUL5–RBX2 forming CRL–RBR E3–E3 complexes were tested by adding their corresponding ARIH-family RBR E3 partner (ARIH1 or ARIH2, preactivated mutant versions, see Methods for details). **g**, Deneddylation assays as in **f**, but with SENP8 deneddylating enzyme. **a**–**g**, Gels, scans and blots are representatives from n = 2 technically independent experiments.

Furthermore, XL–MS data for WT CUL9–RBX1 displayed cross-links that can only be rationalized in the context of the hexameric structure (Extended Data Fig. 8c). One of the most abundant cross-links (K188 with K188 in the ARM1 dimerization interface) was absent in XL–MS for the 'monomer' mutant (Extended Data Fig. 8c,d and Supplementary Tables 4 and 5). It is noteworthy that unneddylated CUL9$^{K1881R}$–RBX1 remained hexameric (Fig. 4b).

Purification of dimeric complexes allowed re-evaluation of oligomerization status of endogenous CUL9. Comparing migration in sucrose gradients suggests that some cellular CUL9 is hexameric, while a smaller fraction aligns with a dimer (Fig. 1f and Extended Data Fig. 8e).

We also examined the effects of the mutants on ubiquitylation activity. The ARM1 dimer and monomer were substantially impaired at ubiquitylating the substrates TP53 and APEX2. The cullin dimer retained ubiquitylation activity toward the substrates, although with a distinct banding pattern for TP53 observed by SDS–PAGE (Fig. 4c and Extended Data Fig. 8f). Thus, the CRL–RBR assembly, maintained in the

cullin dimer, is critical, while hexamerization may enable additional catalytic geometries in which the various active sites access different substrate lysines. Future studies will be required to determine how formation of dimeric versus hexameric assemblies is regulated, and their potential functional differences.

## Distinct neddylation pathway for CUL9

Given the essential role of neddylation for CUL9–RBX1 ubiquitylation activity, we confirmed that NEDD8 modification of endogenous and overexpressed CUL9 in U2OS cells depends on the NEDD8 E1 (NAE). Treatment with the inhibitor MLN4924 (ref. 55) eliminated CUL9 neddylation (Fig. 5a). The NEDD8 modification depends on the structurally observed neddylation site, K1881, and is independent of CUL9's catalytic C2249 (Fig. 5b).

We next asked which of the two NEDD8 E2s, UBE2M or UBE2F, is capable of the modification. Knockdown of UBE2F in U2OS cells substantially reduced NEDD8 modification at the molecular weight corresponding to

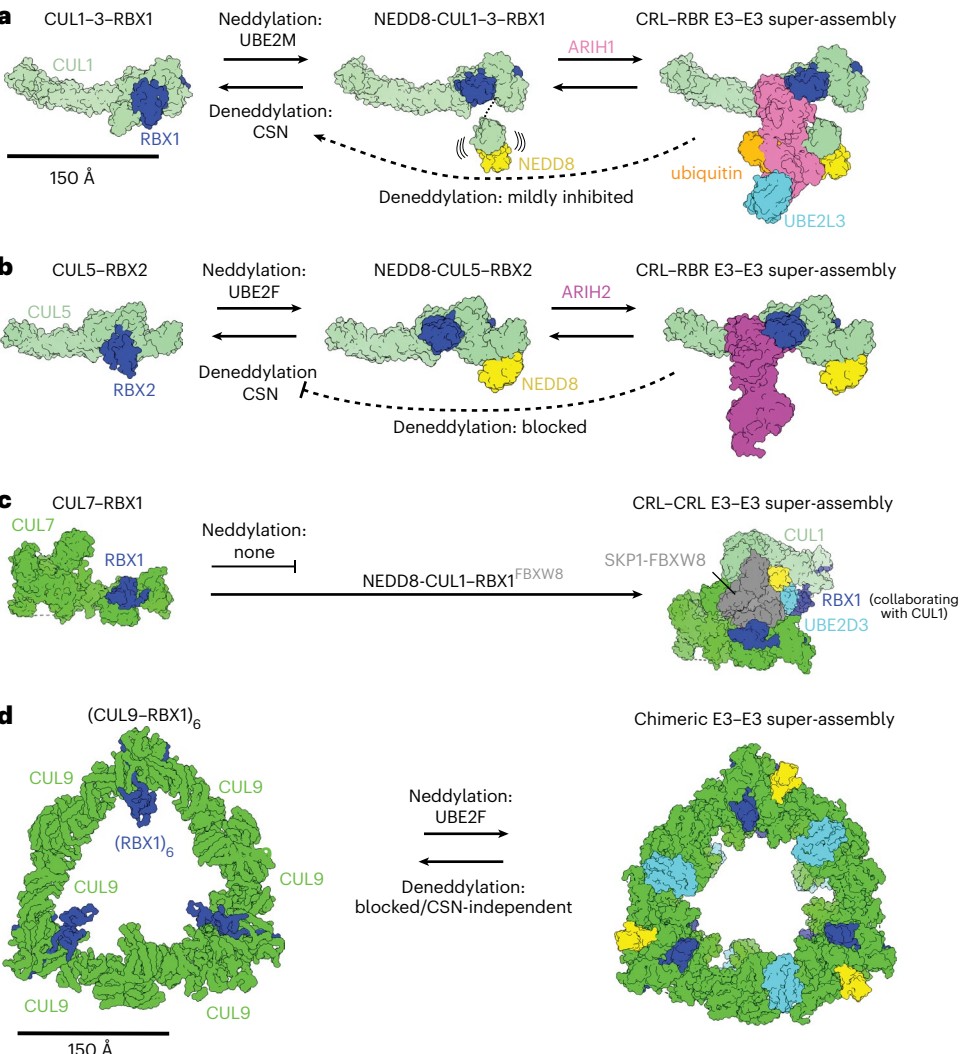

**Fig. 6 | Cullin neddylation, assembly with ARIH-family E3s and effects on deneddylation.** Schematic displaying CRL structures (surface representation) in their unneddylated, (neddylated) states and as E3–E3 super-assembly representation. **a**, The left shows that in cells, cullins 1–3 (represented by (PDB 1LDJ) are preferentially neddylated via UBE2M. The deneddylase CSN removes NEDD8. The center shows neddylated cullins 1–3, with a flexible NEDD8-WHB unit, can form E3–E3 super-assemblies with ARIH1 (represented by PDB 7B5L, showing only NEDD8-linked CUL1–RBX1). The right shows active neddylated CUL1–RBX1-ARIH1 E3–E3 super assembly in conformation for ubiquitin transfer from UBE2L3 to ARIH1 (PDB 7B5L, substrate receptor complex not shown). The active conformation of the E3–E3 assembly is poised to inhibit CSN-mediated deneddylation. **b**, The left shows unneddylated CUL5 in complex with RBX2 (PDB 6V9I), which is neddylated by UBE2F, and deneddylated by CSN. The center shows that neddylated CUL5–RBX2 assembles with ARIH2 to form an E3–E3 super assembly (PDB 7ONI, ARIH2 not shown). The right shows active CUL5–RBX2-ARIH2 E3–E3 super assembly (PDB 7ONI). The E3–E3 assembly performs ubiquitylation and blocks CSN-mediated deneddylation. **c**, The left shows CUL7 forms a complex with RBX1 but is not neddylated as canonical CRLs (PDB 7Z8B, SKP1-FBXW8 not shown). On the right, together with neddylated CUL1–RBX1 and SKP1-FBXW8, CUL7–RBX1 forms an active CRL–CRL E3–E3 super assembly (PDB 7Z8B). **d**, In this study, hexameric CUL9–RBX1 is neddylated by UBE2F, and with chimeric E3–E3 ligase activity encompassed within the CUL9 polypeptide. Recombinant WT CUL9–RBX1 was not deneddylated in vitro by CSN or SENP8, possibly restricted by NEDD8 binding to the built-in ARIH-RBR domain. Even after deletion of the protective ARIH-RBR domain, deneddylation was CSN-independent.

CUL9, but there was little effect of knocking down UBE2M (Fig. 5c). This was unexpected, because although UBE2F can modify both RBX1 and RBX2-associated cullins in vitro, CUL5 is the only cullin known at present to rely on UBE2F (refs. 31–33). We sought to confirm UBE2F-dependent modification of CUL9–RBX1 in vitro. However, we detected little modification of CUL9–RBX1 under the conditions of our assay, despite robust neddylation of CUL1–RBX1 and CUL5–RBX2 in side-by-side control reactions (Fig. 5d,e). Examination of the structure showed that the arrangement of CUL9's WHB domain and RBX1 RING would require reorientation to achieve the conformation for neddylation[46]. Moreover, the ARIH-RBR element appears to protect NEDD8 linked to protomer B (Fig. 2c). Deletion of the ARIH-RBR domain dramatically reduced the NEDD8 modification of CUL9 expressed in mammalian cells (Fig. 5b),

which allowed examining neddylation in vitro. CUL9$^{\Delta ARIH-RBR}$–RBX1 was robustly modified on K1881 when incubated with NAE, MgATP, fluorescent NEDD8 (*NEDD8) and UBE2F, whereas no modification was observed with UBE2M (Fig. 5d,e and Extended Data Fig. 9a).

We were intrigued by the striking modification of CUL9$^{\Delta ARIH-RBR}$, because this is paralogous throughout its length to CUL7. Yet, to date, CUL7 has not been found to be neddylated. A previous structure showed the basis for neddylation of an RBX1-bound fragment of CUL1 spanning from the 4HB domain through the WHB domain[46]. Indeed, replacing this portion of CUL7 with the corresponding sequence from CUL9–without the ARIH-RBR element–and vice versa, showed this region (with RBX1) is necessary and sufficient for neddylation by UBE2F (Extended Data Fig. 9b).

Canonical CRLs are regulated through cycles of neddylation and deneddylation. The latter is catalyzed by the cullin-specific deneddylase CSN, confirmed by effects of pharmacological inhibition of CSN[56] in U2OS cells (Extended Data Fig. 9c). However, immunoblotting for NEDD8 suggested that CUL9 modification was unchanged on CSN inhibition (Extended Data Fig. 9c). We performed experiments with purified components to test our hypothesis that CUL9's ARIH-family RBR E3 element inhibits deneddylation. In control reactions, CUL1–RBX1 and CUL5–RBX2 were readily deneddylated by CSN. We tested effects of adding their ARIH-family RBR counterparts, using ARIH1 and ARIH2 mutants that enhance binding to their neddylated CRL partners[10,26]. Under our assay conditions CUL1–RBX1 deneddylation was mildly inhibited by ARIH1, and CUL5–RBX2 deneddylation was completely prevented when ARIH2 was present (Fig. 5f). However, incubation with CSN did not overtly affect NEDD8 modification of either WT CUL9 or in vitro neddylated CUL9[ΔARIH-RBR]–RBX1. Structural modeling CSN on CUL9 by docking homologous regions of a previous CSN-CUL2 structure[57] showed major clashing between CSN and neddylated CUL9 (Fig. 5f and Extended Data Fig. 9d,e). We thus assayed the only other NEDD8-specific protease, SENP8. SENP8 is known to catalyze NEDD8 maturation and deconjugation from noncullin proteins[58–61]. In the control reactions, SENP8 indeed failed to remove NEDD8 from CUL1 or CUL5, yet it efficiently deconjugated NEDD8 from CUL9[ΔARIH-RBR] (Fig. 5g). Although future studies will be required to determine whether SENP8 or another enzyme deneddylates CUL9 in vivo, our finding that the NEDD8 modification on WT CUL9 remained recalcitrant to deneddylation further hints toward RBR-based self-protection of the modified cullin domain. The ARIH-RBR element's protection of NEDD8 on CUL9 may be a means of preserving the active state.

## Discussion

Our structural and biochemical studies reveal the unprecedented assembly, ubiquitin ligase activity and neddylation of the noncanonical cullin–RING complex, CUL9–RBX1. The CUL9 structure distinctly encompasses both neddylated CRL and RBR E3 functionalities within a single polypeptide, in a giant triangular, hexameric self-assembly. Mechanistic insights were provided through a subset of our recombinant CUL9–RBX1 having been neddylated in human cells, copurifying with an E2 and ubiquitylating TP53 and APEX2 in vitro (Figs. 2 and 3).

CUL9–RBX1 displays a unique combination of CRL E3 and ARIH-family RBR E3 properties. The breadth of differences from canonical E3s is further expanded by the distinct unneddylated and neddylated conformations. NEDD8 linkage redirects CUL9's WHB domain from interactions restraining RBX1's RING domain, and directly binds CUL9's RBR region in the active conformation.

It is exciting to find an atypical pathway mediating CUL9 neddylation (Fig. 6). Previously, UBE2F had only been found essential for neddylating RBX2-bound CUL5 (refs. 31–33). UBE2F-dependent regulation of CUL9 and CUL5 is presumably related to these proteins emerging late in evolution[20,22]. Structural modeling of UBE2F on RBX1's RING in the unneddylated protomer A suggests that neddylation requires yet another CUL9–RBX1 conformation[31,46]. Thus, interesting questions for the future are: what factor or factors are missing from our recombinant system to drive neddylation? What steers CUL9–RBX1 to achieve the conformation for neddylation? And, does neddylation occur in a hexamer, or in monomeric or dimeric precursors before self-assembly? CUL9's neddylation status may be regulated differently from canonical CRLs, where substrates inhibit CSN-mediated deneddylation[35,36,62]. Although we cannot definitively exclude the possibility that CSN deneddylates CUL9, we did not observe such activity in vitro, nor an effect of CSN inhibition on cellular CUL9. Rather, CUL9 deneddylation was only detected with the promiscuous deneddylase SENP8, and only after removing CUL9's ARIH-RBR element. Although it remains unknown what could toggle CUL9's ARIH-RBR element's grip on NEDD8, our structural data showed that the Ariadne and RBR

domains can adopt different positions relative to the CUL9 scaffold (Fig. 2e,g). Our data also raise the possibility that neddylated CUL9 could undergo autodegradation in the absence of substrate—as has been observed for canonical CRLs[63,64]—because it performs autoubiquitylation (Fig. 3b), and inhibiting neddylation slightly increased cellular CUL9 (Fig. 5a).

Finally, this work establishes a structural framework for understanding giant CUL9–RBX1 assemblies. Distinct CUL9 domains—the CPH and DOC domain—are required for ubiquitylation of distinct substrates (Fig. 3h). These properties are reminiscent of the recruitment of different substrates and regulators to distinct domains in another giant E3 (HUWE1) and other E3s forming large oligomers (BIRC6, UBR5 and the yeast GID–human CTLH complex)[2–4,65–71]. Oligomerization may allow intermolecular stabilization of catalytic assemblies, multiple catalytic geometries, avid substrate binding and formation of alternative assemblies with different functions. Indeed, we found that abrogating CUL9–RBX1 oligomerization alters substrate ubiquitylation (Fig. 4c and Extended Data Fig. 8f). Furthermore, the cullin homo-oligomerization interface of CUL9–RBX1 could be mirrored in an alternative dimeric assembly with CUL7–RBX1. CUL9–RBX1 and CUL7–RBX1 use homologous domains to achieve TP53 E3 ligase activity in different ways, CUL9 via its ARIH-RBR E3 element (Fig. 2g,h) and CUL7 through recruiting FBXW8-SKP1 and neddylated CUL1–RBX1 (ref. 21) (Fig. 6). In addition to their both regulating cytoplasmic TP53 (refs. 11,72), CUL7 and CUL9 bind each other in a manner that restrains ubiquitylation activity[15]. Although we were unable to obtain pure CUL9–CUL7 complexes, superimposing the previous CUL7–RBX1 structure[21] onto one CUL9–RBX1 protomer suggests these two proteins could potentially form an unneddylated mixed cullin dimer. We speculate that differences in CUL7's ARM1 domain could prevent CUL9–RBX1 from forming the hexameric assembly, while the cullin dimer with CUL9–RBX1 would prevent CUL7 from binding FBXW8-SKP1 to achieve E3 ligase activity. Given the multidomain natures of CUL7 and CUL9—with CUL9 also showing ARIH-family RBR E3 ligase activity—we anticipate many fascinating variations on these giant CRLs and other E3–E3 complexes executing ubiquitylation.

## Online content

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

## Methods

### Construct design, protein expression and purification

All proteins in this study are of human origin.

**CUL9–RBX1 and variants: construct design and expression.** Complementary DNA encoding full-length CUL9 (residues 1–2517) and RBX1 (residue 5 to C terminus) were cloned into pEG vectors individually, with CUL9 carrying a N-terminal TwinStrep-tag and consequent 3C protease cleavage site. Subsequently, gene expression cassettes were combined into a single pBIG1a vector[73], which was used for bacmid generation from emBacY *Escherichia coli*. After introducing the bacmid into Sf9 insect cells (bought from Thermo Fisher, identifier no. 11496015) through transfection, the baculovirus was amplified and increased up to the third passage (P3). The resulting baculovirus-containing supernatant was then sterile filtered and used for infection of HEK293S cells. HEK293S GnTI⁻ were bought from the American Type Culture Collection (ATCC) (identifier CRL-3022). HEK293S cells were grown to a density of around $3 \times 10^6$ cells per ml, infected with 10% (v/v) of virus and incubated for 16 h at 37 °C. Next, 10 mM sodium butyrate was added, the temperature was decreased to 30 °C and finally the cells were collected after 48 h (ref. [74]).

CUL9 variants, either lacking selected residues or domains, or containing point mutations, were expressed by simultaneously infecting HEK293S cells with two separate baculoviruses. One baculovirus encoded the specific TwinStrep-tagged CUL9 variant, while the other carried the RBX1 gene.

To study the effects of selected CUL9 domains and sequences, the following residues were deleted in the listed CUL9 variants by replacing them with a GSGSGSGS linker:

CUL9^ΔCPH: 354–460
CUL9^ΔARM9: 599–924
CUL9^ΔARM3: 948–1105
CUL9^ΔDOC: 1167–1296
CUL9^ΔRING1: 2057–2142
CUL9^ARM1dimer: 1652–1690
CUL9^monomer: 1652–1690, and carries the following point mutations: R125A and Y152A

CUL9 variants were also obtained by truncation of N- and/or C-terminal sequences. The following variants are truncated at the indicated residue(s):

CUL9^ΔARIH-RBR: 1–1978
CUL9^K1881R-ΔARIH-RBR: 1–1978 and carries the K1881R point mutation
CUL9^ARIH-RBR: 1979–2517

Chimeric versions between CUL9 and CUL7 were generated by combining selected regions of both proteins:

CUL7^CUL9-chimera: CUL7 residues 1–1208 (SBD to CR3) + CUL9 1538–1978 (4HB to WHB)
CUL9^CUL7-chimera: CUL9 residues 1–1537 (SBD to CR3) + CUL7 1209–1698 (4HB to WHB)

All CUL9 variants were coexpressed with RBX1 except for CUL9^ARIH-RBR as this only encompasses the ARIH-RBR sequence, which does not bind RBX1.

**Protein purification of CUL9–RBX1 and variants.** HEK293S cells were collected by centrifugation and resuspended in lysis buffer (50 mM HEPES pH 7.5, 200 mM NaCl, 1 mM dithiothreitol (DTT), 1× cOmplete Protease Inhibitor, Roche). Subsequently, cells were lysed via sonication and centrifuged for 30 min at 20,000*g*. The protein-containing supernatant was incubated with Strep-Tactin resin for 30 min at 4 °C. After transferring the resin to gravity flow columns, five washing steps were performed using wash buffer (25 mM HEPES pH 7.5, 200 mM NaCl, 1 mM DTT). The protein was eluted with 25 mM HEPES pH 7.5, 200 mM NaCl, 1 mM DTT, 2.5 mM Desthiobiotin. Next, Strep-tagged fusion protein was cleaved by incubation with 3C protease (molar ratio 1:50 protease:protein) for 2–3 h

at room temperature and purified by size-exclusion chromatography on a Superose 6 Increase 10/300 GL column (GE Healthcare) using 25 mM HEPES pH 7.5, 150 mM NaCl, 1 mM DTT. The elution volume of the protein complex was close to the Superose 6 void volume and indicated formation of a larger assembly.

**Expression and purification of TP53.** His-lipoyl domain-tagged TP53 was expressed in *E. coli* BL21 Rosetta and induced with isopropyl beta-ᴅ-thiogalactoside (IPTG) (0.5 mM) at an optical density of 0.6–0.8 and expression continued at 18 °C overnight. *E. coli* cells were disrupted via sonication and cell lysate subjected to centrifugation at 20,000*g* for 30 min. Target protein-containing supernatant was subjected to immobilized metal affinity chromatography. After elution with imidazole, fusion protein was dialyzed overnight at 4 °C with tobacco etch virus (TEV) protease (50 mM Tris pH 7.5, 150 mM NaCl, 5 mM DTT, 1:50 molar ratio target protease:protein). Size-exclusion chromatography in 25 mM HEPES pH 7.5, 150 mM NaCl, 1 mM DTT on a Superose 6 Increase 10/300 GL column (GE Healthcare) was used to purify tetrameric TP53 away from tag and protease. The purified TP53 was used as a substrate in ubiquitylation assays.

**Expression and purification of CSN.** CSN constructs for insect cell expression were generated using the biGBac system[73]. Here, CSN3 features a C-terminal 3C cleavage site, succeeded by a 3× Strep-tag, while CSN5 carries an N-terminal 6× His-tag with subsequent 3C site. All other subunits including CSN1, CSN2, CSN4, CSN6, CSN7b and CSN8 were untagged. These CSN subunits 1–8 were coexpressed from a single baculovirus in *Trichoplusia ni* High-Five insect cells. Insect cells were collected and lysed as described for HEK293S cells and the protein complex-containing supernatant was incubated with Strep-Tactin beads. After washing and elution, immobilized metal affinity chromatography (Ni-NTA) was performed and affinity tags were cleaved overnight while dialyzing in 50 mM HEPES pH 7.5, 150 mM NaCl, 1 mM DTT, 3C protease in 1:50 molar ratio. Finally, size-exclusion chromatography was carried out to buffer exchange into in 25 mM HEPES pH 7.5, 150 mM NaCl, 1 mM DTT.

**Expression and purification of GST-tagged proteins: ubiquitin, NEDD8, UBA1, E2 enzymes, CUL1–RBX1, CUL5–RBX2, ARIH1, ARIH2 and SENP8.** Full-length human CUL1 and CUL5 were coexpressed with their respective RING protein GST-TEV-RBX1 (residue 5 to C terminus) or GST-TEV-RBX2 (residue 5 to C terminus) in *Trichoplusia ni* High-Five insect cells. GST-TEV-UBA1 was also expressed in insect cells. Full-length UBE2A, UBE2B, UBE2C, UBE2D1, UBE2D2, UBE2D3, UBE2D4, UBE2E1, UBE2E2, UBE2E3, UBE2F, UBE2G1, UBE2G2, UBE2H, UBE2I, UBE2J1, UBE2J2, UBE2K, UBE2L3, UBE2M, UBE2N, UBE2Q2, UBE2R1, UBE2R2, UBE2S, UBE2T, UBE2V1, UBE2V2, SENP8, NAE1-UBA3, ARIH1, ARIH2 and mutant versions (the so-called OPEN mutants that mutationally release autoinhibitory Ariadne-Rcat domain interactions, F430A E431A E503A for ARIH1, and L381A E382A E455A for ARIH2) cloned into pGEX-4T1 vectors were expressed as GST-TEV fusion proteins in *E. coli* Rosetta 2 (DE3). Expression was induced with IPTG (0.1 mM for ARIH1 and ARIH2, 0.5 mM for E2s, Ubiquitin, NEDD8 and SENP8) at an optical density of 0.6–0.8. For ARIH1 and ARIH2, 0.1 mM ZnCl₂ was added to the TB medium. Expression continued overnight at 18 °C for all proteins. Cell lysates containing GST-fusion proteins were subjected to disruption via sonication. Following centrifugation, protein-containing supernatant was then allowed to incubate with glutathione sepharose beads. The beads were washed several times with wash buffer (50 mM Tris pH 7.5, 250 mM NaCl, 1 mM DTT). Cleavage with TEV protease was performed on beads overnight. After elution of cleaved protein from the column with wash buffer, the target protein was subjected to ion exchange and size-exclusion chromatography in a final buffer of 25 mM HEPES pH 7.5, 150 mM NaCl, 0.5 mM tris(2-carboxyethyl)phosphine (TCEP)[7,9,10,46,75].

**Expression and purification of APEX2.** C-terminal 3× FLAG-tagged APEX2 was expressed analogously to CUL9 in HEK293S cells. Cell lysate was incubated for 1 h with anti-FLAG M2-affinity gel, washed five times with buffer (25 mM HEPES pH 7.5, 150 mM NaCl) and eluted with the same buffer including 100 ng ml$^{-1}$ FLAG-peptide. Eluted protein was subjected to ion exchange, concentrated and buffer exchanged into 25 mM HEPES pH 7.5, 150 mM NaCl, 1 mM DTT before usage as a substrate in ubiquitylation assays.

**Expression and purification of ubiquitin.** WT ubiquitin was produced in *E. coli* Rosetta 2 (DE3) cells and subsequently purified without the use of a tag. The purification process involved a glacial acetic acid purification step[76], followed by ion exchange using an S-column and subsequent size-exclusion chromatography. In short, acetic acid was slowly added to the bacterial lysate until a pH of ~4.5 was reached. This precipitated most proteins other than ubiquitin. After dialysis into 25 mM sodium acetate pH 4.5, 100 mM NaCl, the dialyzed ubiquitin was centrifuged and cleared supernatant was subjected to ion exchange chromatography on a S-column, followed by size-exclusion chromatography on an SD75 10/300 GL column (GE Healthcare) into 25 mM HEPES pH 7.5, 150 mM NaCl, 1 mM DTT to yield WT, tagless ubiquitin.

## Cryo-EM

**Sample preparation and data collection for CUL9–RBX1.** After size-exclusion chromatography, peak fractions of CUL9–RBX1 were pooled and concentrated to 5 mg ml$^{-1}$. The protein complex was then cross-linked in batch with 0.05% (v/v) Glutaraldehyde for 10 min at room temperature, followed by the addition of and incubation with 50 mM Tris pH 7.5 for another 5 min to quench the cross-linking reaction. Shortly before plunging, fluorinated Fos-Choline-8 (Anagrade) was added to the protein sample at a final concentration of 1.5 mM. This was essential to overcome preferred orientation of the sample. Subsequently, holey carbon grids (Quantifoil, R1.2/1.3, 200 mesh) were glow discharged, and 3 µl of CUL9–RBX1 was applied to the grid at 95% humidity and 4 °C using a Vitrobot Mark IV (Thermo) and plunge-frozen into liquid ethane (blot force 3, blot time 3 s). After several screening datasets to identify the ideal compromise between ice thickness and particle density, high-resolution data were collected on a Titan Krios transmission electron microscope (TEM), equipped with a post-GIF Gatan K3 Summit direct electron detector in counting mode. Datasets were collected using SerialEM (v.3.8.0-b5) and FEI EPU (v.2.7.0). Videos were collected at a nominal magnification of ×105,000, equaling 0.8512 Å/pixel at the specimen level. The target defocus ranged between −0.7 and −2.8 µm and the total dose of ~60 e/Å$^2$ was distributed over 40 frames.

**Processing of CUL9–RBX1 cryo-EM data.** Motion-correction and dose weighting were performed using RELION v.3.1 (ref. [77]) and the contrast transfer function (CTF) was estimated using CTFFIND-4.1 (ref. [78]). Particles were picked using Gautomatch (v.0.56) (K. Zhang, MRC Laboratory of Molecular Biology) with reference-free blob-based picking. Then, 16,800 micrographs with a maximum resolution estimate better than 5 Å were imported into RELION v.3.1 (ref. [77]), from which ~1.21 million particles were extracted applying 5.25× binning. These were subjected to several rounds of 3D classification, followed by initial model generation and 3D classification. After 3D classification with and without masking, several sets of particles for the hexamer, the cullin dimer or the E2-density map were re-extracted at full pixel size. Subsequently, masked 3D auto-refinement, CTF-refinement and particle polishing resulted in maps with resolutions of less than 5 Å. RELION[77] postprocessing and DeepEMhancer (v.2020.09.07)[79] were used for sharpening of the final maps. A higher-resolution map of the inactive (unneddylated) cullin dimer complex was obtained by performing symmetry expansion following the application of C3 symmetry during the previous refinement. During 3D classification, particles with the neddylated conformation were excluded. Local refinement resulted

in a 3.37 Å map of the cullin dimer, where both protomers are in the inactive (unneddylated) conformation. Unfortunately, symmetry expansion while selecting for the neddylated (active) conformation did not yield substantially more particles or result in map improvement.

**Model building and refinement.** Coordinates were built for three complexes, in the following sequence: the mixed cullin dimer between neddylated and unneddylated CUL9–RBX1 using the map shown in Fig. 2c, a hexamer using the map shown in Fig. 1b, and an unneddylated CUL9–RBX1 dimer using the map shown in Extended Data Fig. 3c.

The cullin dimer structure comprising neddylated and unneddylated CUL9–RBX1 also contains an E2 and was built as follows. A structural model of CUL9, predicted by Alphafold2 (ref. [39]), was split into several domains and segments that were fit into the cryo-EM map using Chimera (v.1.13.1)[80]. Models for most domains could be initially docked in the well-defined secondary structure, with the exception of the CUL9 CPH, ARM9, DOC and Rcat domains, which could not be placed in the map. The CUL9 Ariadne domain was clearly resolved in both the neddylated and unneddylated protomers, albeit in different relative orientations. The remaining regions of the CUL9 ARIH-RBR element were only resolved in the neddylated protomer. RBX1 was built based on the CRL7$^{FBXW8}$ structure[21]. The NEDD8 linked to CUL9 was built based on the structure representing ubiquitin transfer from UBE2L3 to ARIH1 bound to a neddylated CUL1-based CRL[9]. That structure also provided coordinates for UBE2L3 docked into the density for an E2 bound to the CUL9 ARIH-RBR element RING1 domain. UBE2L3 was used for E2 in the structure based on ubiquitylation assays, AP–MS, XL–MS and isothermal calibration (ITC) binding experiments. Ultimately, the model was completed by iterative cycles of manual rebuilding and refinement using Coot (v.0.8.9.1)[81], alternating with real-space refinements with Phenix.refine (v.1.17.1)[82]. For lower resolution parts of the map, side-chains were removed, including wholesale removal of side-chains across the CUL9 ARM1-3 domains, NEDD8, and the E2 (UBE2L3). For the RBX1 RING domain, the side-chain placement was maintained from the starting model from the complex with CUL7 (ref. [21]). The coordinates for this cullin dimer complex (comprising unneddylated CUL9–RBX1 and E2-bound neddylated CUL9–RBX1) served as the starting model for the other two structures.

The cryo-EM map of the full hexamer did not allow placement of side-chains but most domains other than the ARIH-RBR element and NEDD8 were clearly visible on a secondary structure level. The structure of the dimeric complex between unneddylated and neddylated CUL9–RBX1—without the ARIH-RBR element and NEDD8—was fit into the hexameric cryo-EM map three times using Chimera (v.1.13.1)[80]. Clear density was observed for the three unneddylated protomers, while the CUL9 WHB domain RBX1 RING domain from the alternating protomers were poorly resolved and thus these regions were removed from the coordinate file. Side-chains and remaining unresolved segments were removed in Coot[81], and the structure was finalized by rigid body refinement with Phenix.refine (v.1.17.1)[82]. It seems likely that relatively lower resolution of the CUL9 WHB domain and RBX1 RING domain in some protomers results from intrinsic conformational heterogeneity and/or a mixture of neddylated and unneddylated complexes. To represent both versions in a single hexamer, the position of the RING domain from the neddylated CUL9–RBX1 complex was shown for alternating protomers in Fig. 1c.

To obtain the structure of the unneddylated cullin dimer, the coordinates for the unneddylated CUL9 protomer, bound to the N-terminal strand from RBX1, from the dimer described above were fit using Chimera (v.1.13.1)[80] into the cryo-EM map obtained by symmetry expansion. Side-chains were remodeled using Coot (v.0.8.9.1)[81], followed by real-space refinement with Phenix.refine (v.1.17.1)[82]. The RBX1 RING domain from the published complex with CUL7 (ref. [21]) was wholesale docked into remaining density, and the final model was polished by rigid body refinement with Phenix.refine (v.1.17.1)[82].

**Sample preparation for CUL9$^{\Delta CPH}$–RBX1, CUL9$^{\Delta ARM9}$–RBX1, CUL9$^{\Delta ARIH-RBR}$–RBX1 and CUL9$^{\Delta DOC}$–RBX1 and data collection.** CUL9$^{\Delta CPH}$–RBX1, CUL9$^{\Delta ARM9}$–RBX1, CUL9$^{\Delta ARIH-RBR}$–RBX1 and CUL9$^{\Delta DOC}$–RBX1 were purified following the same protocol as for WT CUL9–RBX1, concentrated to 3 mg ml$^{-1}$ and snap frozen in liquid nitrogen for storage. Before plunging, samples were thawed on ice and centrifuged for 10 min at 4 °C, 14,000$g$. Cross-linking and plunging was performed as for WT CUL9–RBX1. Datasets were collected either on an Arctica TEM equipped with a Falcon III electron detector in linear mode, or on a Glacios TEM equipped with K2 Summit direct detector in counting mode. Videos were captured using the Arctica TEM with a nominal magnification of ×73,000, resulting in a pixel size of 1.997 Å/pixel at the specimen level. Alternatively, videos were recorded on the Glacios TEM with a nominal magnification of ×22,000, yielding a pixel size of 1.885 Å/pixel at the specimen level. The intended defocus spanned from −1.2 to −3.3 μm, and the cumulative exposure, approximately 60 electrons per Å², was distributed across 40 frames.

**Processing of cryo-EM data for CUL9$^{\Delta CPH}$–RBX1, CUL9$^{\Delta ARM9}$–RBX1, CUL9$^{\Delta ARIH-RBR}$–RBX1 and CUL9$^{\Delta DOC}$–RBX1.** Motion-correction and dose weighting were performed using RELION 4.0 (ref. 77) and the CTF was estimated using CTFFIND-4.1 (ref. 78). Particles were picked using Gautomatch (v.0.56) (K. Zhang, MRC Laboratory of Molecular Biology) with reference-free blob-based picking. All datasets used the hexameric CUL9–RBX1 template as reference for initial 3D classification without application of symmetry, followed by several iterations of 3D refinement and alignment-free 3D classification. Finally, clean particle sets were unbinned and refined, followed by PostProcessing in RELION[77]. Cryo-EM maps were analyzed in ChimeraX (v.1.2.5).

## Assays to assess ubiquitylation, neddylation and deneddylation

**Fluorescent labeling of ubiquitin and NEDD8.** Ubiquitin and NEDD8 were expressed with an additional N-terminal cysteine. This cysteine was ultimately used to label the proteins fluorescently. After size-exclusion chromatography into 25 mM HEPES, pH 7.5, 150 mM NaCl and 5 mM DTT, protein was desalted twice with Zeba Spin Desalting columns to remove DTT as it would be interfering in the reaction with the maleimide. Next, fluoresceine-5-Maleimide (dissolved in anhydrous dimethylsulfoxide (DMSO)) was incubated with ubiquitin with a tenfold molar excess. The overall concentration of DMSO did not surpass 5% in the reaction. This mixture was then incubated at room temperature for 2 h before the addition of 10 mM DTT to halt the reaction. The reactions were desalted to eliminate any remaining unreacted maleimide. Subsequently, the reaction mixture underwent two rounds of size-exclusion chromatography into 25 mM HEPES (pH 7.5), 150 mM NaCl and 1 mM DTT, yielding fluorescent ubiquitin (*Ub) or NEDD8 (termed *N8 or *NEDD8 in figures and text).

**Ubiquitylation assays.** All ubiquitylation reactions were performed in a multi-turnover format. Assays screen CUL9–RBX1-dependent ubiquitylation activity with a range of E2 enzymes, and compare activities of WT CUL9–RBX1 and E2 UBE2L3 versus variants toward TP53 and APEX2 substrates. Ubiquitylation assays were prepared by mixing 0.25 μM UBA1, 1 μM E2, 1 μM E3 (WT or variant), 15 μM *Ub, with or without 1 μM substrate (TP53 or APEX2) in 25 mM HEPES pH 7.5, 100 mM NaCl, 2.5 mM MgCl$_2$. The reaction was started by addition of 2.5 mM ATP, incubated at room temperature and quenched at indicated time points with SDS–PAGE sample buffer. SDS–PAGE gels were imaged with an Amersham Typhoon Imager (Cy2 channel) to visualize fluorescently labeled ubiquitin.

**Neddylation assays.** Posttranslational modification of cullins with NEDD8 on their respective WHB domains was assayed in the same format as for ubiquitylation reactions. For this purpose, 0.5 μM NAE (NAE1-UBA3), 1 μM E2 (either UBE2F or UBE2M), 1 μM cullin (CUL1–RBX1, CUL5–RBX2 or CUL9–RBX1 or variants thereof) and 5 μM *NEDD8 were mixed in 25 mM HEPES pH 7.5, 150 mM NaCl, 2.5 mM MgCl$_2$ and reaction was started by addition of 2.5 mM ATP (final concentration). The reactions were quenched at the indicated time points with SDS–PAGE sample buffer, subjected to SDS–PAGE and analyzed using an Amersham Typhoon Imager (Cy2 channel) to visualize fluorescent NEDD8. Alternatively, if neddylation reaction were to be subsequently used to study deneddylation of neddylated cullin by either CSN or SENP8, reactions were quenched with either 10 mM DTT (in the case of CSN) or 15 mM EDTA (for SENP8).

**Deneddylation assays.** NEDD8 modification on canonical cullins is specifically removed by the deneddylase CSN. In contrast, SENP8 is a deneddylase responsible for proteolytic cleavage of pro-NEDD8, hyper-neddylated cullins and other proteins. To investigate whether NEDD8 modification on CUL9 could be removed by either of the deneddylases, 2 μM SENP8 or 0.1 μM CSN was added to the quenched neddylation reactions. The deneddylation reactions were stopped at the indicated time points by addition of SDS–PAGE sample buffer and SDS–PAGE gels were imaged with an Amersham Typhoon Imager (Cy2 channel) to visualize Fluorescein-labeled NEDD8.

## Biochemical and biophysical characterization of CUL9–RBX1 and variants

**Mass photometry.** WT CUL9–RBX1 was analyzed using mass photometry to estimate size and oligomeric state. Calibration was performed by using a protein mixture containing a variety of molecular masses, including Aprotinin, Ribonuclease A, Carbonic anhydrase, Ovalbumin, Conalbumin and Blue dextran, all present at a final concentration of approximately 20 nM for each component. WT CUL9–RBX1 was measured in a final concentration of 50 nM in 25 mM HEPES pH 7.5, 150 mM NaCl, 0.5 mM TCEP. Data were collected over 50 frames and 30 s on a Refeyn TwoMP mass photometer using Refeyn AcquireMP v.2.3.0. Data were analyzed with the Refeyn DiscoverMP v.2.3.0 software.

**SEC–MALS.** In addition to mass photometry, SEC–MALS analysis was performed to estimate the molecular weight of the CUL9–RBX1 complex. For this purpose, 70 μl of purified protein at 3 mg ml$^{-1}$ was loaded onto a Superdex 200 10/300 GL column (GE Healthcare) connected to a DAWN8 + TREOS MALLS detector and Optilab rEX differential refractometer (Wyatt Technologies). Each run was performed at a flow rate of 1 ml min$^{-1}$ in 25 mM HEPES pH 7.5, 150 mM NaCl, 1 mM TCEP at room temperature. Molecular-weight calculations were performed with ASTRA software v.5.3 (Wyatt Technologies).

**Size-exclusion chromatography of CUL9–RBX1 variants.** Size-exclusion chromatography was used to examine the oligomeric status of CUL9–RBX1 and CUL9–RBX1 variants containing mutations designed to disrupt the dimerization interfaces (Fig. 4b and Extended Data Fig. 1a). To establish a reference, a size standard mixture (Bio-Rad), containing thyroglobulin, γ-globulin, ovalbumin, myoglobin and vitamin B12 was loaded onto a Superose 6, 5/150GL column (GE). Subsequently, a 50 μl sample of 1.5 μM WT or variant CUL9–RBX1 was loaded onto the Superose 6, 5/150GL column (GE). The gel filtration buffer contained 25 mM HEPES pH 7.5, 150 mM NaCl, 1 mM DTT for all runs.

**Sucrose gradients.** To study endogenous CUL9 assemblies, sucrose gradient fractionation was performed. For this, 1 mg of total protein cell lysate was loaded onto a continuous 5–40% sucrose gradient (weight and volume in 25 mM HEPES 7.5, 150 mM NaCl, 1 mM DTT, 1 mM EDTA, 0.05% TWEEN and 1× cOmplete Protease Inhibitor Mix), which was generated via gradient maker (Biocomp Gradient Master 108). Samples were centrifuged in an ultracentrifuge (Thermo Scientific Sorvall WX+ Ultracentrifuge) equipped with a SW60Ti rotor at 160,000$g$ for 16 h

at 4 °C. Fourteen 300 µl fractions were collected from the top of the gradient, separated by SDS–PAGE and followed by immunoblotting using indicated antibodies. The blots were developed using Clarity Western ECL Substrate (Bio-Rad, catalog no. 16640474) and imaged using an Amersham Imager 600 (GE Lifesciences). Bio-Rad's Gel Filtration Standard and purified WT hexameric CUL9–RBX1, as well as dimeric CUL9^(R125A Y152A)–RBX1 were run for comparison. Endogenous CUL9 samples were run in triplicate and distribution of CUL9 protein over the fraction was plotted, normalized to the total CUL9 protein amount in all fractions.

**ITC analysis.** ITC measurements were performed on a MicroCal PEAQ-ITC (Malvern) at 25 °C with a setting of 19 × 2 µl injections. CUL9^ARIH, UBE2D2, UBE2D3 or UBE2L3 were all dialyzed into dialysis buffer (25 mM HEPES pH 7.5, 150 mM NaCl, and 0.5 mM TCEP) before analysis. For measurements, the syringe contained a concentration of E2 enzyme at 300–500 µM and the cell contained CUL9^(ARIH-RBR) at 25 µM. The heats of dilution for diluting E2s into measurement buffer were subtracted from binding experiments before curve fitting. Manufactured supplied software was used to fit the data to a single-site binding model and to determine the stoichiometry ($N$), the molar reaction enthalpy $\Delta H$, the entropy change $\Delta S$ and the association constant $K_a$. The dissociation constant, $K_D$, was calculated from $1/K_a$.

### Cell culture and cell treatments
**U2OS cell culture.** U2OS cells (ATCC HTB-96)[17] were maintained in McCoy's 5A medium (Gibco) supplemented with 10% fetal calf serum (Gibco), 100 U ml⁻¹ penicillin, 0.1 mg ml⁻¹ streptomycin (Gibco) at 37 °C, 5% $CO_2$. For better detection, CUL9 was subcloned into pcDNA5 FRT/TO vector with HA tag at the C terminus. To ensure this was the only CUL9 present, U2OS ΔCUL9 cells[17] were transiently transfected using Lipofectamine 3000 (Thermo Fisher) according to the manufacturer's protocol and incubated for 48–96 h at 37 °C before performing further analysis.

**Immunoblot analysis of cell lysates.** The cells were gathered by centrifugation at 360$g$, washed once with ice-cold 1× PBS, and resuspended in lysis buffer (25 mM HEPES 7.5, 150 mM NaCl, 1 mM DTT, 1 mM EDTA, 0.05% TWEEN and 1× cOmplete Protease Inhibitor Mix (Roche)), supplemented with 0.01% TWEEN and incubated on ice for 10 min. Cells were homogenized by douncing ten times. The obtained lysates were cleared by centrifugation at 23,000$g$ for 30 min at 4 °C, and protein concentration was determined by Micro BCA-Protein Assay (Thermo Scientific, catalog no. 23235). For immunoblot analysis, lysates were denatured with SDS sample buffer, boiled at 95 °C for 5 min, separated on SDS–PAGE and proteins were visualized by immunoblotting using indicated primary antibodies: NEDD8 (CST, 2745), CUL9 specific antibody was a gift from A. Alpi, Vinculin (Abcam, catalog no. ab129002) and β-Actin (CST, catalog no. 4967). All primary antibodies in this paper were used at a final concentration of 1 µg ml⁻¹.

**U2OS cell treatment with MLN4924 and CSN5i-3.** MLN4924 inhibits NAE enzyme and thus neddylation in cells[55]. CSN5i-3 is an inhibitor that targets the catalytic CSN5 subunit of the CSN, and prevents deneddylation of canonical cullins[56]. To test how both inhibitors affect CUL9, U2OS parental and ΔCUL9 knock-out cells were treated with either 0.5 µM MLN4924 (also known as Pevonedistat, Selleckchem, S7109) or 3 µM CSN5i-3 (MCE, HY-112134). Cell viability and confluency were carefully monitored throughout the incubation period, ensuring the confluency remained between 40 and 90%. Control cells were treated with DMSO. After 24 h of drug treatment, cell lines were transiently transfected with HA-tagged CUL9 using Lipofectamine 3000 (Thermo Fisher) following the manufacturer's protocol and incubated for further 24 h before cell lysis and immunoblot analysis.

**siRNA knockdown of UBE2F and UBE2M.** U2OS cells were seeded at a density of 30–40% cells per well, ensuring approximately 70–80% confluence on the day of transfection. Small-interfering RNAs (siRNAs) were obtained from Dharmacon. For siRNA knockdown, the cells were transfected with 40 pmol siRNA targeting UBE2F (CAAGUAAACUGAAGCGUGA, AUGACUACAUCAAACGUUA, CAAUAAGAUACCCGCUACA, CUGAAGUUCCCGAUGCGUA, catalog numbers J-009081-09, J-009081-10, J-009081-11 and J-009081-12), UBE2M (GAAAUAGGGUUGGCGCAUA, AAGCCAGUCCUUACGAUAA, UUAAGGUGGGCCAGGGUUA, GAUGAGGGCUUCUACAAGA, J-004348-05, J-004348-06, J-004348-07 and J-004348-08) or nontargeting (UGGUUUACAUGUCGACUAA) using RNAi Max (Thermo Fisher, 13778075) according to manufacturer's protocol. The transfected cells were incubated at 37 °C for 48 h, after which the cells were lysed and knockdown efficiency was assessed by immunoblotting.

**Co-immunoprecipitation in U2OS cells.** HA-tagged proteins were captured from 1 mg total cell lysate using anti-HA affinity matrix (Pierce, catalog no. 88836) overnight at 4 °C. All immunoprecipitation reactions were washed in lysis buffer, and immunoadsorbed proteins were eluted by boiling in reducing SDS sample buffer, separated by SDS–PAGE followed by immunoblotting using indicated antibodies.

### Mass spectrometry analyses
**XL–MS.** *Sample preparation.* Purified WT CUL9–RBX1 or the monomeric variant were cross-linked at a concentration of 4 µM protein complex with 2 mM bis(sulfosuccinimidyl)suberate for 20 min at room temperature. Cross-linking was quenched by adding 50 mM Tris-HCl pH 7.5 (final concentration) and incubated for 5 min. Cross-linked proteins were denatured, reduced and alkylated by addition of 4 M urea, 40 mM 2-cloroacetamide (Sigma-Aldrich) and 10 mM TCEP (Thermo Fisher Scientific) in 50 mM Tris-HCl. After incubation for 20 min at 37 °C, the samples were diluted 1:3 with mass spectrometry grade water (VWR) and proteins were digested overnight at 37 °C by addition of 0.5 µg of LysC and 1 µg of trypsin (Promega). Thereafter, the solution was acidified with trifluoroacetic acid (TFA) (Merck) to a final concentration of 1%, followed by desalting of the peptides using Sep-Pak C18 1cc vacuum cartridges (Waters).

*Data acquisition LC–MS analysis.* Peptides were dissolved in buffer A (0.1% formic acid) and 1/20 of the peptides were analyzed by liquid chromatography with tandem mass spectrometry (LC–MS/MS) comprising an Easy-nLC 1200 (Thermo Fisher Scientific) coupled to an Exploris 480 or a QExactive HF mass spectrometer (Thermo Fisher Scientific). Peptides were separated within 60 min on a 30 cm analytical column (inner diameter 75 µm; packed in-house with ReproSil-Pur C18-AQ 1.9 µm beads, Dr. Maisch GmbH) using a gradient of buffer A to buffer B (80% acetonitrile (ACN), 0.1% FA). The mass spectrometer was operated in data-dependent mode and specialized settings for the data acquisition of cross-linked peptides were set: we have used higher-energy C-trap dissociation with normalized collision energy values of 19, 27 and 35, and we have excluded charge state 2 from being fragmented to enrich the fragmentation scans for cross-linked peptide precursors.

*Data processing XL–MS.* The acquired raw data were processed using Proteome Discoverer (v.2.5.0.400) with the XlinkX/PD nodes integrated[83]. The database search was performed against a FASTA file containing the sequences of the proteins under investigation. Disuccinimidyl suberate was set as a cross-linker. Cysteine carbamidomethylation was set as fixed modification and methionine oxidation and protein N-terminal acetylation were set as dynamic modifications. Trypsin/P was specified as protease and up to two missed cleavages were allowed. Identifications were only accepted with a minimal score of 40 and a minimal delta score of 4. Filtering at 1% false-discovery rate at the cross-link spectrum match (CSM) and cross-link level was applied. The data were analyzed with cross-link analyzer v.1.1.4.

**AP–MS.** *Expression and pulldown*. CUL9–RBX1 and variants were expressed as described above but in triplicates and the Strep-pulldown elutions were subsequently processed for mass spectrometry.

*Sample preparation*. For the reduction and alkylation of the proteins, 100 µl of SDC buffer (1% sodium deoxycholate, 40 mM 2-chloroacetamide (Sigma-Aldrich), 10 mM TCEP (Pierce™, Thermo Fisher Scientific) in 100 mM Tris-HCl, pH 8.0) was added and the proteins were incubated for 20 min at 37 °C. The samples were diluted 1:2 with water and digestion proceeded overnight at 37 °C by addition of 0.5 µg of trypsin (Promega). The solution of peptides was then acidified with TFA (Merck) to a final concentration of 1% followed by purification via SCX StageTips. Samples were vacuum dried and resuspended in 12 µl of buffer A (0.1% formic acid (Roth) in mass spectrometry grade water (VWR)).

*LC–MS/MS data acquisition*. Here, 800 ng of the desalted peptide mixture was separated on an analytical column (30 cm, 75 µm inner diameter, packed in-house with ReproSil-Pur C18-AQ 1.9 µm beads, Dr. Maisch GmbH) by an Easy-nLC 1200 (Thermo Fisher Scientific) at a flow rate of 250 nl min$^{-1}$ while heating the column to 60 °C. The LC was coupled to a QExactive HF mass spectrometer (Thermo Fisher Scientific). As a LC-gradient, the following steps were programmed with increasing addition of buffer B (80% ACN, 0.1% formic acid): linear increase from 7 to 30%B over 60 min, followed by a linear increase to 60%B over 15 min, then followed by a linear increase to 95%B and finally, the percentage of buffer B was maintained at 95% for another 5 min.

The mass spectrometer was operated in a data-dependent mode with survey scans from 300 to 1,650 $m/z$ (resolution of 60,000 at $m/z = 200$), and up to ten of the top precursors were selected and fragmented using higher-energy collisional dissociation (with a normalized collision energy of value of 28). The MS2 spectra were recorded at a resolution of 15,000 (at $m/z = 200$). AGC target for MS1 and MS2 scans were set to $3 \times 10^6$ and $1 \times 10^5$, respectively, within a maximum injection time of 100 and 60 ms for MS1 and MS2 scans, respectively.

*Data analysis*. Raw data were processed using the MaxQuant computational platform (v.2.2.0.0)[84] with standard settings applied. The peak list was searched against the Human UniProt database (SwissProt and TrEMBL) with an allowed precursor mass deviation of 4.5 ppm and an allowed fragment mass deviation of 20 ppm. Cysteine carbamidomethylation was set as static modification, and methionine oxidation and N-terminal acetylation as variable modifications. The match between-run option was enabled, and proteins were quantified across samples using the label-free quantification algorithm in MaxQuant generating label-free quantification intensities.

**Mass spectrometric analysis of ubiquitylation sites on CUL9–RBX1 and TP53.** *Sample preparation*. Here, 4 µM CUL9–RBX1 was incubated with 0.2 µM UBA1, 4 µM UBE2L3, 40 µM WT ubiquitin and 2.5 mM MgATP with or without TP53 for 30 min at room temperature. The reactions were quenched with 10 mM DTT and 6 µg of total protein amount was alkylated, reduced and digested simultaneously using 1 M urea in 50 mM ABC with 10 mM TCEP, 40 mM 2-chloracetamide and 0.5 µg of trypsin (Sigma-Aldrich) at 37 °C overnight with agitation (1,500 rpm) on an Eppendorf Thermomixer C. SDB-RPS (Empore) StageTips were used for peptide desalting. In brief, peptides were diluted using a 1:10 ratio (peptide, 1% TFA in isopropanol), loaded to StageTips and washed with 200 µl of 1% TFA in isopropanol and then with 0.2% TFA/2% ACN twice. Peptide elution was done using 75 µl of 80% ACN/1.25% NH$_4$OH. Samples were then dried using a SpeedVac centrifuge (Concentrator Plus; Eppendorf) for 1 h at 30 °C and subsequently resuspended 0.2% TFA/2%. Finally, 50 ng of peptides were injected into LC–MS/MS.

*Data-dependent acquisition LC–MS analysis*. For LC–MS/MS analysis, we used the following setup: 50 cm reversed phase column (75 µm inner diameter, packed in-house with ReproSil-Pur C18-AQ 1.9 µm resin), a homemade oven that maintained a column temperature constant at 50 °C, an EASY-nLC 1200 system (Thermo Fisher Scientific) connected online to the mass spectrometer (Orbitrap Exploris 480, Thermo Fisher Scientific) via a nano-electrospray source. For peptide separation we used a binary buffer system (buffer A, 0.1% formic acid and buffer B, 80% ACN, 0.1% formic acid). Peptides were eluted using a 60 min gradient with a constant flow rate of 300 nl min$^{-1}$. The gradient starts at 3% buffer B and increases to 8% after 35 min, 36% after 40 min, 45% after 44 min and 95% after 48 min until it stays constant until 52 min and decreases to 5% buffer B after 60 min. The following settings were used for mass spectrometry data acquisition: data-dependent acquisition mode with a full scan range of 250–1,350 $m/z$, 60,000 resolution, $3 \times 10^6$ automatic gain control (AGC), 20 ms maximum injection time and 28 higher-energy collision dissociation. Every survey scan was followed by 12 data-dependent acquisition scans with a 30,000 resolution, a $1 \times 10^6$ AGC and a 110 ms maximum injection time.

*Data processing and bioinformatics analysis*. Raw files were process using MaxQuant v.1.6.2.10 (ref. [84]). For the search we used a human UniProt FASTA file with 42,347 entries. The digestion mode was set to trypsin/P with a maximum of two missed cleavage sites and maximum and minimum peptide lengths of 25 and 8, respectively. Variable modifications were set to oxidation (M), acetyl (Protein N-term) and GlyGly (K) and fixed modification were set to carbamidomethylation (C). Match between run was enabled. The bioinformatics analyses were done using Python v.3.5.5 with the following packages: numpy v.1.21.5, and pandas v.1.4.2.

**Reporting summary**

Further information on research design is available in the Nature Portfolio Reporting Summary linked to this article.

## Data availability

The atomic coordinates and cryo-EM maps have been deposited in the RCSB Protein Data Bank (PDB) with accession codes PDB ID 8Q7H (focused neddylated and unneddylated cullin dimer), PDB ID 8Q7E (hexameric assembly), PDB ID 8RHZ (unneddylated cullin dimer built-in symmetry expanded map) and the Electron Microscopy Data Bank with codes EMD-18216 (focused neddylated and unneddylated cullin dimer), EMD-18214 (hexameric assembly), EMD-19179 (unneddylated cullin dimer symmetry expanded map), EMD-18218 (focused dimeric core), EMD-18217 (focused on E2 density), EMD-18220 (CUL9$^{\Delta CPH}$–RBX1), EMD-18222 (CUL9$^{\Delta ARM9}$–RBX1), EMD-18223 (CUL9$^{\Delta ARIH\text{-}RBR}$–RBX1) and EMD-18221 (CUL9$^{\Delta DOC}$–RBX1). The mass spectrometry data have been deposited to the ProteomeXchange Consortium (http://proteomecentral.proteomexchange.org) via the PRIDE repository with the dataset identifiers PXD047326 and PXD047229. Tables of cross-links are provided as Supplementary Information. Source data are provided with this paper.

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

## Acknowledgements

We thank J. Botsch, L. Hehl, G. Kleiger, M. Kluegel, S. v. Gronau and J. Kellermann for assistance, helpful discussions and/or critical reading of the manuscript. We thank D. Bollschweiler and T. Schäfer of the cryo-EM facility as well as M. Zobawa, C. Basquin and S. Uebel of the biochemistry core facility at Max Planck Institute of Biochemistry. We are grateful to A. Alpi for advice and the anti-CUL9 antibody (Department of Molecular Machines and Signaling, Max Planck Institute of Biochemistry, Martinsried, Germany). Work on this project in the laboratory of B.A.S. was supported by the Max Planck Gesellschaft and has received funding from the European Research Council under the European Union's Horizon 2020 research and innovation program (grant agreement no. 789016-NEDD8Activate). This study was supported by grant no. NIHCA068377 (originally to Y.X., transferred to W. Marzluff).

## Author contributions

D.H.-G., S.K., Y.X. and B.A.S. conceived the project. D.H.-G., L.V.M.H., S.K., J.D., V.B., L.S. and E.M.B., generated protein complexes and/or performed their biochemical characterization. D.H.-G., L.V.M.H., J.D. and E.M.B. performed enzyme assays. D.H.-G. and L.V.M.H. generated cryo-EM samples and collected, processed and refined cryo-EM data. L.V.M.H. built and refined the structures. J.R.P. assisted with cryo-EM data analyses. J.D. performed ITC analysis. I.T.-G. performed cellular studies with assistance from V.B. and reagents from Y.X. D.T.V. and B.S. performed mass spectrometry analyses, supervised by M.M. D.H.-G., L.V.M.H. and B.A.S. analyzed the data and prepared the manuscript with input from the other authors. B.A.S. supervised the project.

## Funding

## Competing interests

B.A.S. is a member of the scientific advisory boards of Biotheryx and Proxygen, and is a co-inventor of intellectual property related to DCN1 inhibitors licensed to Cinsano. Y.X. is Chief Scientific Officer of Cullgen.

## Additional information

**Extended data** is available for this paper at https://doi.org/10.1038/s41594-024-01257-y.

**Correspondence and requests for materials** should be addressed to Brenda A. Schulman.

**Peer review information** *Nature Structural & Molecular Biology* thanks the anonymous reviewers for their contribution to the peer review of this work. Peer reviewer reports are available. Primary Handling Editor: Dimitris Typas was the primary editor on this article and managed its editorial process and peer review in collaboration with the rest of the editorial team.

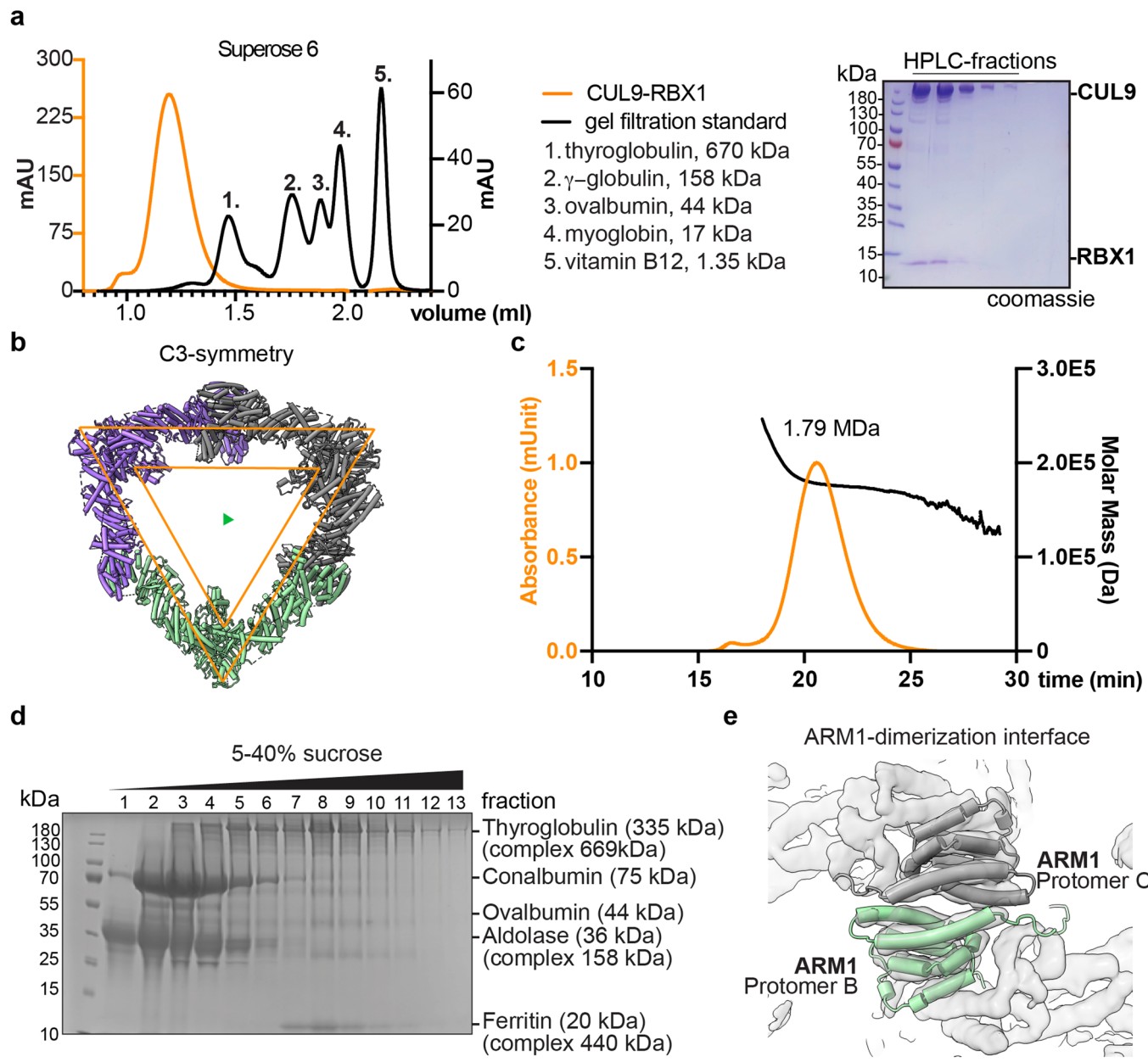

**Extended Data Fig. 1 | Biochemical analysis of the oligomeric assembly formed by CUL9–RBX1. a**, Left: Size-exclusion chromatography profiles of recombinant CUL9-RBX1 (orange) and molecular weight standards (Bio-Rad, black) from a Superose 6, 5/150GL column. Right: Coomassie-stained SDS-PAGE analysis of peak fractions of CUL9-RBX1 (*n* = 2 technically independent experiments). **b**, C3-symmetric hexameric CUL9-RBX1 structure, showing the three constituent cullin dimer subcomplexes in different colors. **c**, Size exclusion chromatography-multiangle light scattering (SEC-MALS) of CUL9-RBX1 confirms hexameric assembly with roughly 1.8 MDa molecular weight. **d**, Coomassie-stained SDS-PAGE analysis of sucrose gradient fractionation of Bio-Rad molecular weight standards (*n* = 2 technically independent experiments). **e**, Close-up of CUL9-RBX1 hexamer structure overlaid with transparent cryo-EM density, focused on the N-terminal ARM1 dimerization interface.

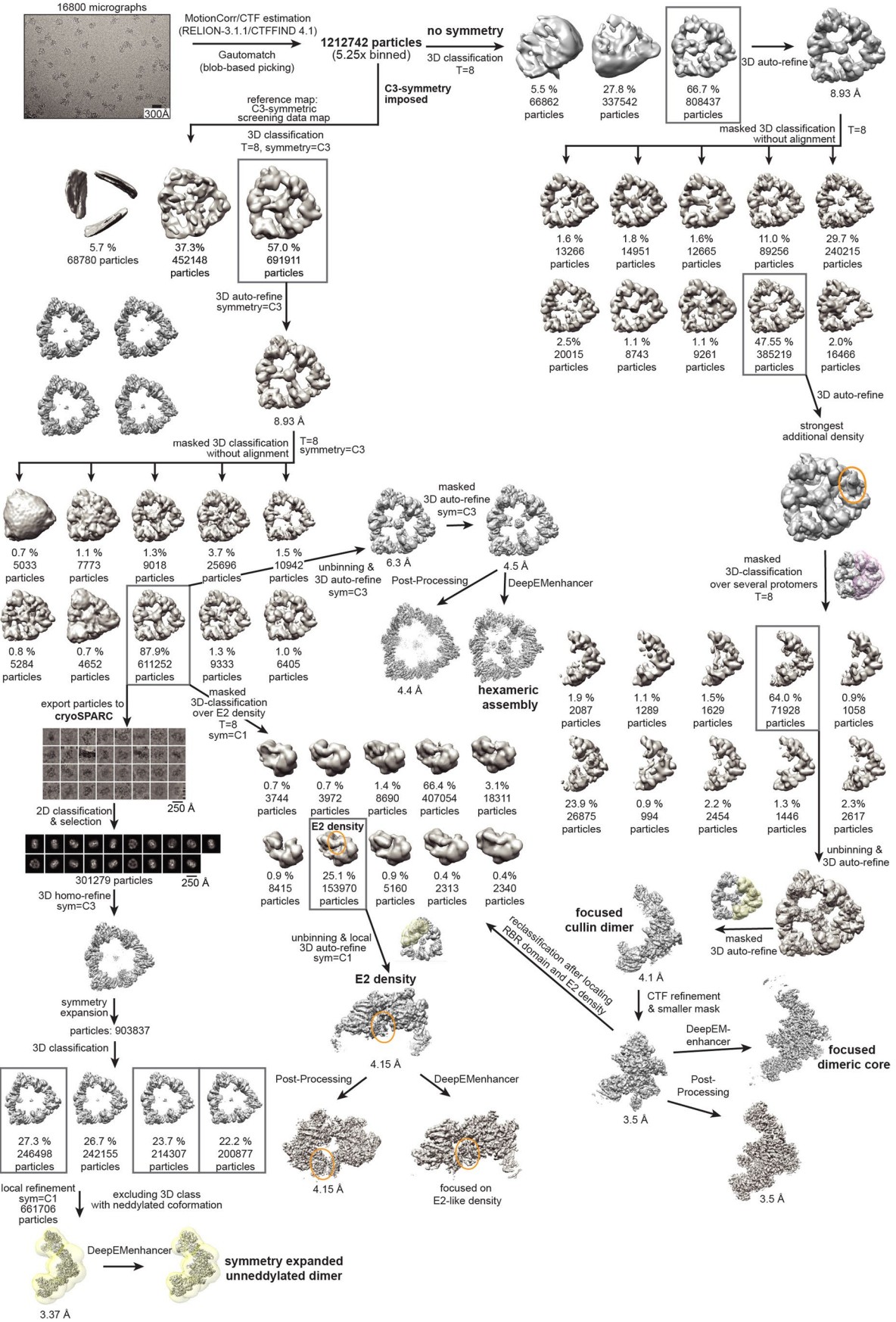

**Extended Data Fig. 2 | See next page for caption.**

**Extended Data Fig. 2 | Cryo-EM processing flowchart for CUL9-RBX1.**
Representative micrographs of CUL9-RBX1 dataset and Cryo-EM image processing flowchart. 16800 micrographs were collected on Titan Krios equipped with a post-GIF Gatan K3 Summit direct electron detector in counting mode. Processing resulted in one map of the full CUL9-RBX1 hexamer at 4.4 Å resolution, one map with a wider mask of the dimer focused on the additional density of the E2 at 4.15 Å, a tighter focused map revealing the ARIH-RBR element at 3.5 Å, one map with a medium tight mask focused on the cullin dimer at 4.1 Å and a map based on symmetry expanded particles of an unneddylated cullin dimer at 3.37 Å resolution.

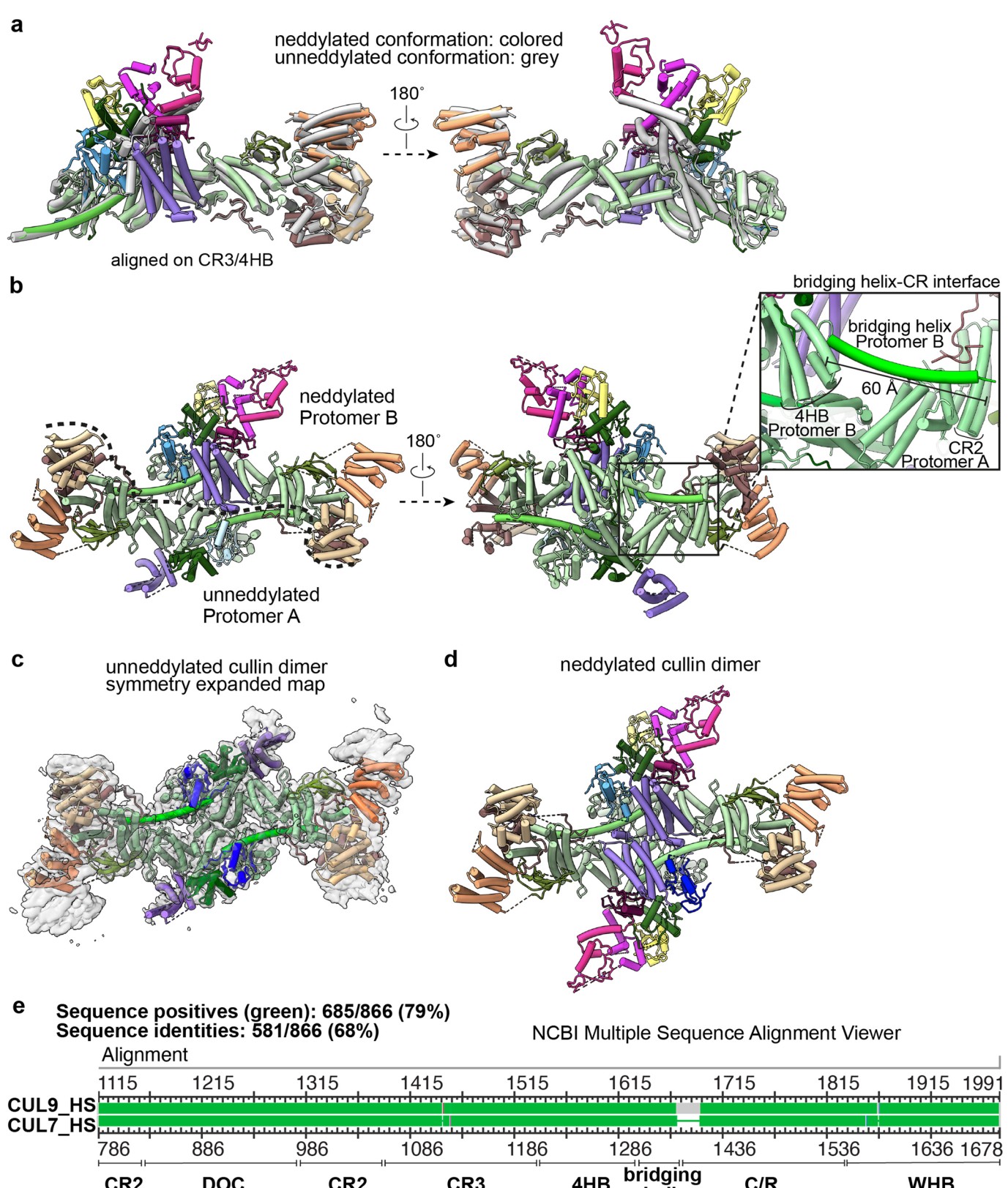

**e** Sequence positives (green): 685/866 (79%)
Sequence identities: 581/866 (68%)

NCBI Multiple Sequence Alignment Viewer

**Extended Data Fig. 3 | Comparison of neddylated and unneddylated conformations of CUL9-RBX1. a**, Neddylated (colored) and unneddylated (grey) protomers of CUL9-RBX1 aligned on CR3-4HB domains. **b**, Structure of dimeric CUL9-RBX1 assembly in two views, domains colored according to Fig. 2a. Upper protomer B is neddylated and has the ARIH-RBR element visible, while the lower protomer A is unneddylated. The close-up on the right visualizes the cullin dimerization interface between the bridging helix of protomer B with the CR domains of protomer A. **c**, Structure of the cullin dimeric subunit

with both Protomers in the unneddylated conformation shown inside the DeepEMhancer map derived from symmetry expansion and focused refinements excluding particles with the neddylated conformation. **d**, Model of a potential cullin dimeric subcomplex with both Protomers neddylated. **e**, Schematic sequence comparison of the cullin-homology domains from CUL9 and CUL7, from NCBI Multiple Sequence Alignment Viewer (https://www.ncbi.nlm.nih.gov/projects/msaviewer/). Sequence positives indicated in green are either identical residues or residues with similar chemical properties.

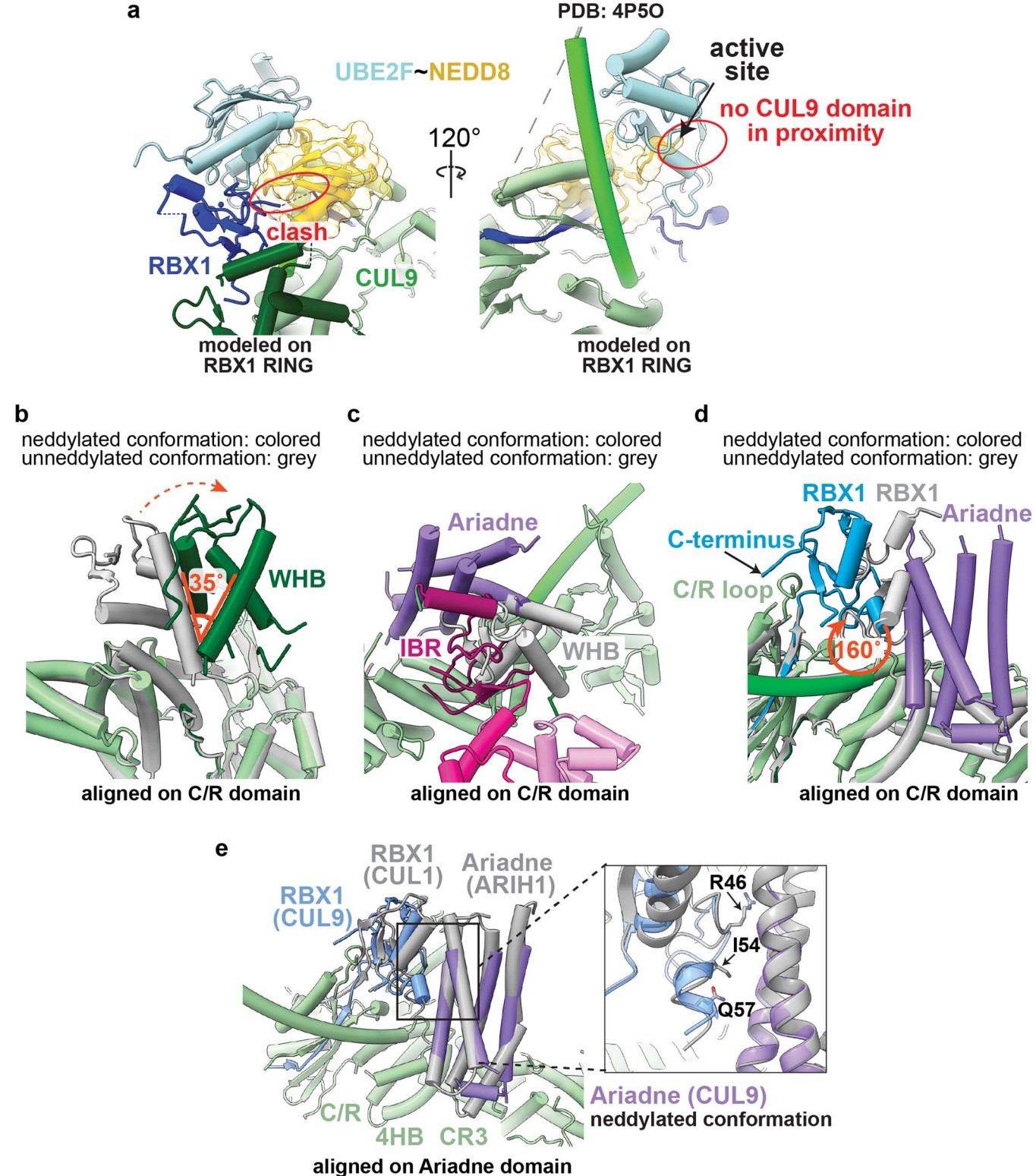

**Extended Data Fig. 4 | Conformational rearrangement of neddylated CUL9-RBX1. a**, Model of the neddylation structure of CUL1 (PDB: 4P5O, with the E2 UBE2M representing similarly structured UBE2F, and DCN1 hidden) aligned with unneddylated CUL9-RBX1 protomer over RBX1 RING domain. **b**, Neddylated (colored) and unneddylated (grey) conformations of CUL9-RBX1 aligned on C/R domain. 35° reorientation of WHB domain highlighted with orange arrow. **c**, After aligning neddylated and unneddylated CUL9-RBX1 over their C/R domains, the WHB domain of unneddylated CUL9-RBX1 (grey) is shown superimposed on neddylated CUL9-RBX1 (colored). The WHB domain from neddylated CUL9 must rearrange due to steric clashing with its Ariadne domain. **d**, RBX1 RING domain reorientation in neddylated (colored) versus unneddylated (grey) CUL9-RBX1. RBX1 RING domain is rotated by 160˚, indicated by orange arrow. **e**, Comparison between binding of the Ariadne domain in the CUL1-ARIH1 E3–E3 super-assembly (PDB: 7B5L, CUL1, SKP1, SKP2, p27, CKSHS1, CDK2, Cyclin A, NEDD8, ubiquitin and UBE2L3 not shown) versus CUL9's intrinsic Ariadne domain and collaborating RBX1. Structures are aligned on the Ariadne domain.

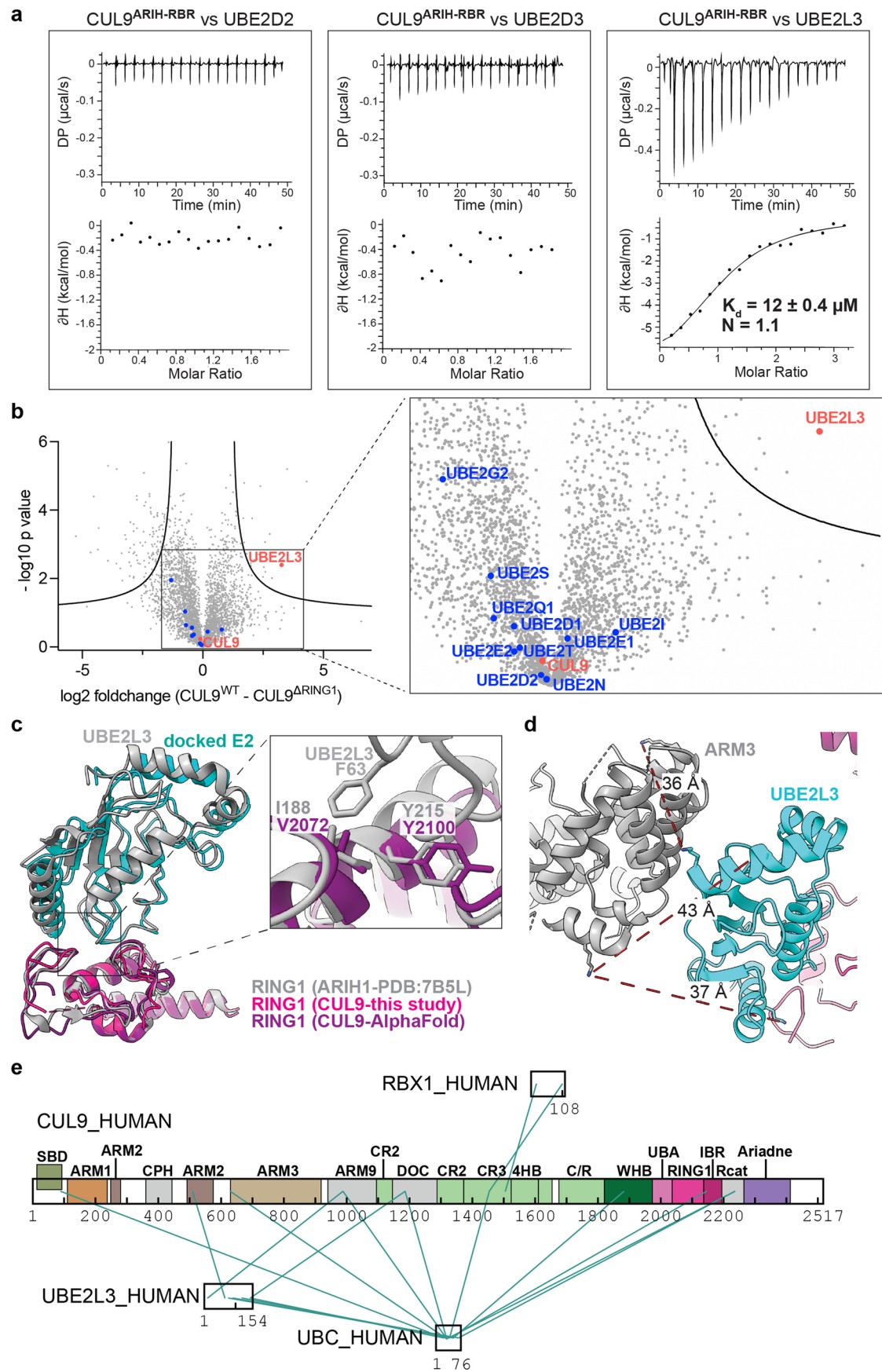

**Extended Data Fig. 5 | See next page for caption.**

**Extended Data Fig. 5 | E2 recruitment by CUL9-RBX1. a**, Isothermal titration calorimetry of CUL9$^{ARIH-RBR}$-RBX1 with UBE2D2, UBE2D3 and UBE2L3. **b**, Comparison of proteins identified by mass spectrometry as interacting with TwinStrep-CUL9-RBX1 versus CUL9$^{\Delta RING1}$-RBX1 (deletion mutant of CUL9's RING1 domain) expressed in HEK293S cells. Volcano plots of p-values (-log10) from two-tailed Student's t tests versus protein abundance (log2) differences. The significance curve was calculated based on a false-discovery-rate-adjusted P = 0.01 and a minimal fold change S0 = 0.6. Proteins above the curve show significant differences between CUL9-RBX1 and CUL9$^{\Delta RING1}$-RBX1. The ubiquitin E2 enzymes identified are highlighted in blue and CUL9 and UBE2L3 highlighted and labeled in red. Data were obtained for each protein from three independent biological replicates. **c**, Overlay of E3 RING1 domain and E2 from ARIH1-UBE2L3 complex (PDB: 7B5L), CUL9-RBX1 RING1 domain and the E2 from this study, and an AlphaFold2 model of the CUL9 RING1 domain aligned on the RING1 domains. **d**, Crosslinks between UBE2L3 and CUL9 ARM3 domain mapped onto the CUL9-RBX1 cullin dimer structure and a modeled neighboring protomer. **e**, Visualization of BS3 cross-linking mass spectrometry analysis of CUL9-RBX1 sample mixed with UBE2L3-ubiquitin. 2D Plots were visualized with XiNET (www.crosslinkviewer.org). Table of crosslinks can be found in Supplementary Table 1.

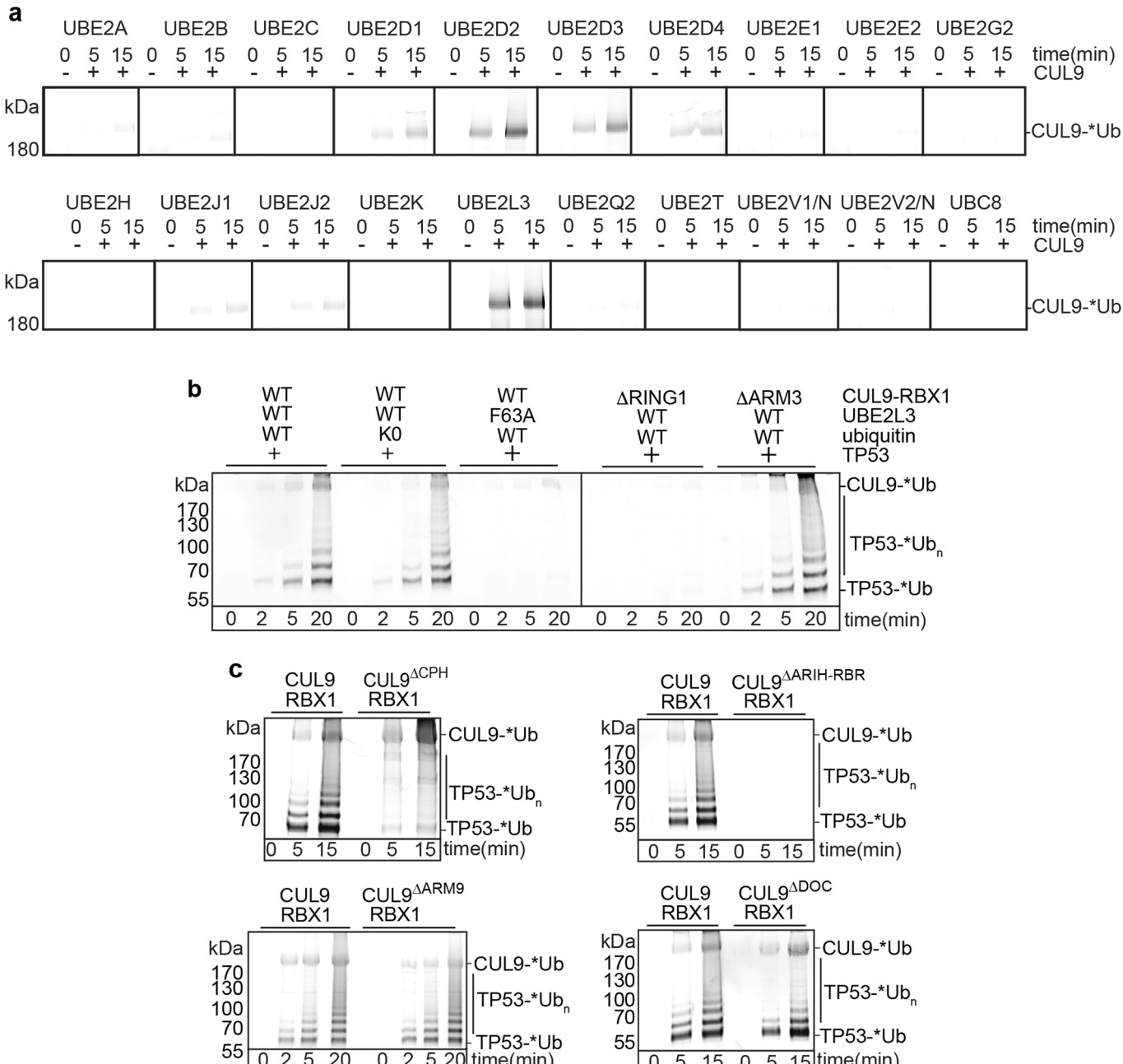

**Extended Data Fig. 6 | Elements of CUL9-RBX1-mediated ubiquitylation.**
**a**, Assays testing a panel of E2 enzymes for capacity to function with CUL9-RBX1, based on CUL9 autoubiquitylation monitored with fluorescent ubiquitin (*Ub). **b**, In vitro ubiquitylation assays testing ubiquitylation activity towards substrate TP53 with UBE2L3, with WT versus lysineless/N-terminally blocked (K0) fluorescent ubiquitin, UBE2L3 Ala substitution for CUL9 RING1-binding F63 residue, and effects of deletion mutant versions of CUL9 lacking the RING1 or ARM3 domains (ΔRING1, ΔDOC). The domains were replaced by a linker of sequence GSGSGSGS. Assays detect fluorescently-labeled ubiquitin (*Ub). **c**, In vitro TP53 ubiquitylation assays comparing modification by recombinant WT CUL9-RBX1 or variants: CUL9-RBX1$^{\Delta CPH}$ in which CPH domain was replaced by a linker of sequence GSGSGSGS, CUL9-RBX1$^{\Delta ARIH\text{-}RBR}$ lacking ARIH-RBR element by truncation at residue 1978, CUL9-RBX1$^{\Delta ARM9}$ in which ARM9 domain was replaced by a linker of sequence GSGSGSGS, and CUL9-RBX1$^{\Delta DOC}$ in which DOC domain was replaced by a linker of sequence GSGSGSGS. Gel scans in all panels are representatives from n = 2 technically independent experiments.

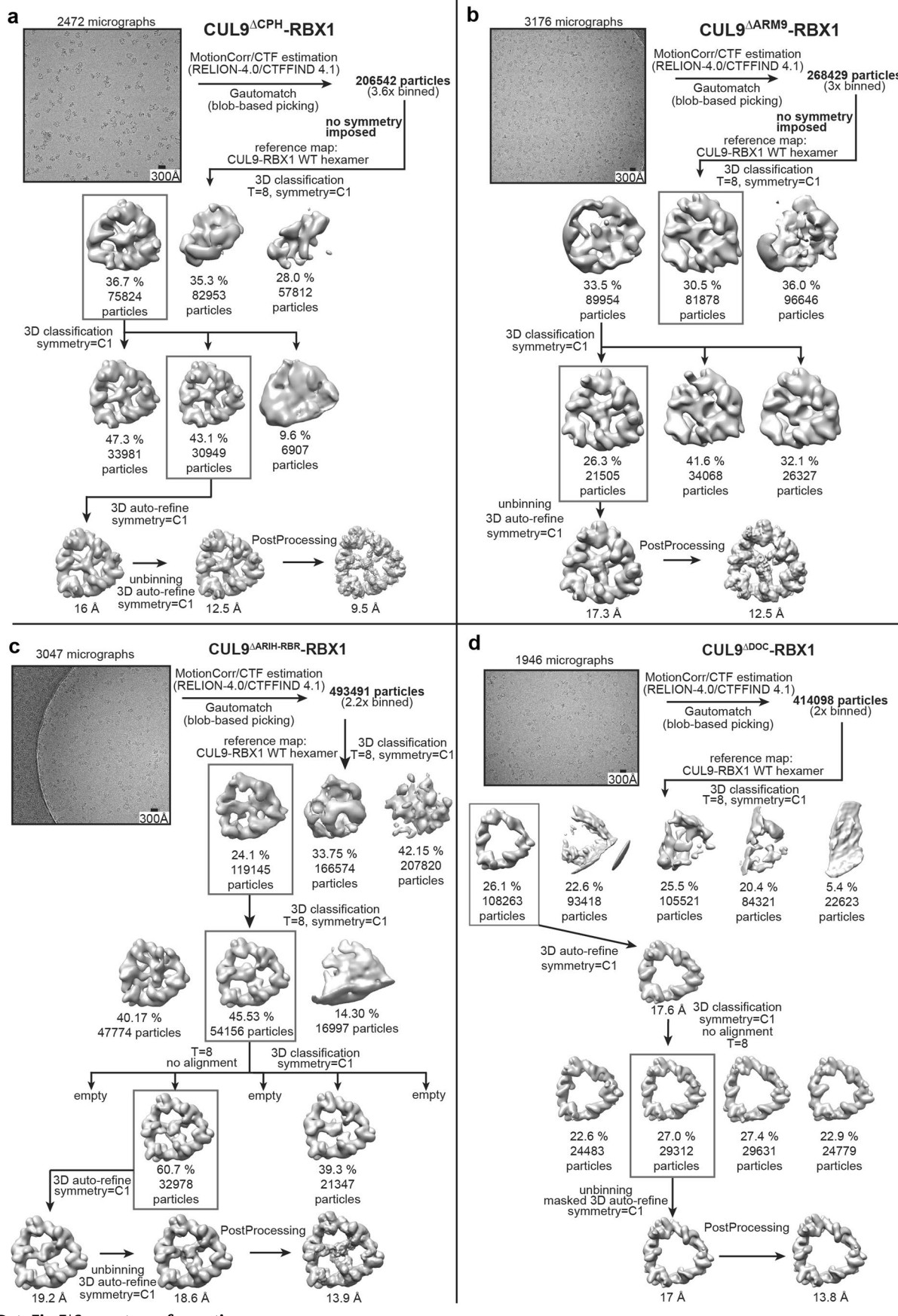

**Extended Data Fig. 7 | See next page for caption.**

**Extended Data Fig. 7 | Cryo-EM processing flowchart for CUL9-RBX1$^{\Delta CPH}$, CUL9-RBX1$^{\Delta ARIH-RBR}$, CUL9-RBX1$^{\Delta ARM9}$ and CUL9-RBX1$^{\Delta DOC}$. a**, Representative micrograph and cryo-EM processing scheme for CUL9-RBX1$^{\Delta CPH}$, in which CPH domain was replaced by a linker of sequence GSGSGSGS. **b**, Representative micrograph and cryo-EM processing scheme for CUL9-RBX1$^{\Delta ARM9}$, in which ARM9 domain was replaced by a linker of sequence GSGSGSGS. **c**, Representative micrograph and cryo-EM processing scheme for CUL9-RBX1$^{\Delta ARIH-RBR}$, a variant lacking ARIH-RBR element by truncation at residue 1978. **d**, Representative micrograph and cryo-EM processing scheme for CUL9-RBX1$^{\Delta DOC}$, in which DOC domain was replaced by a linker of sequence GSGSGSGS.

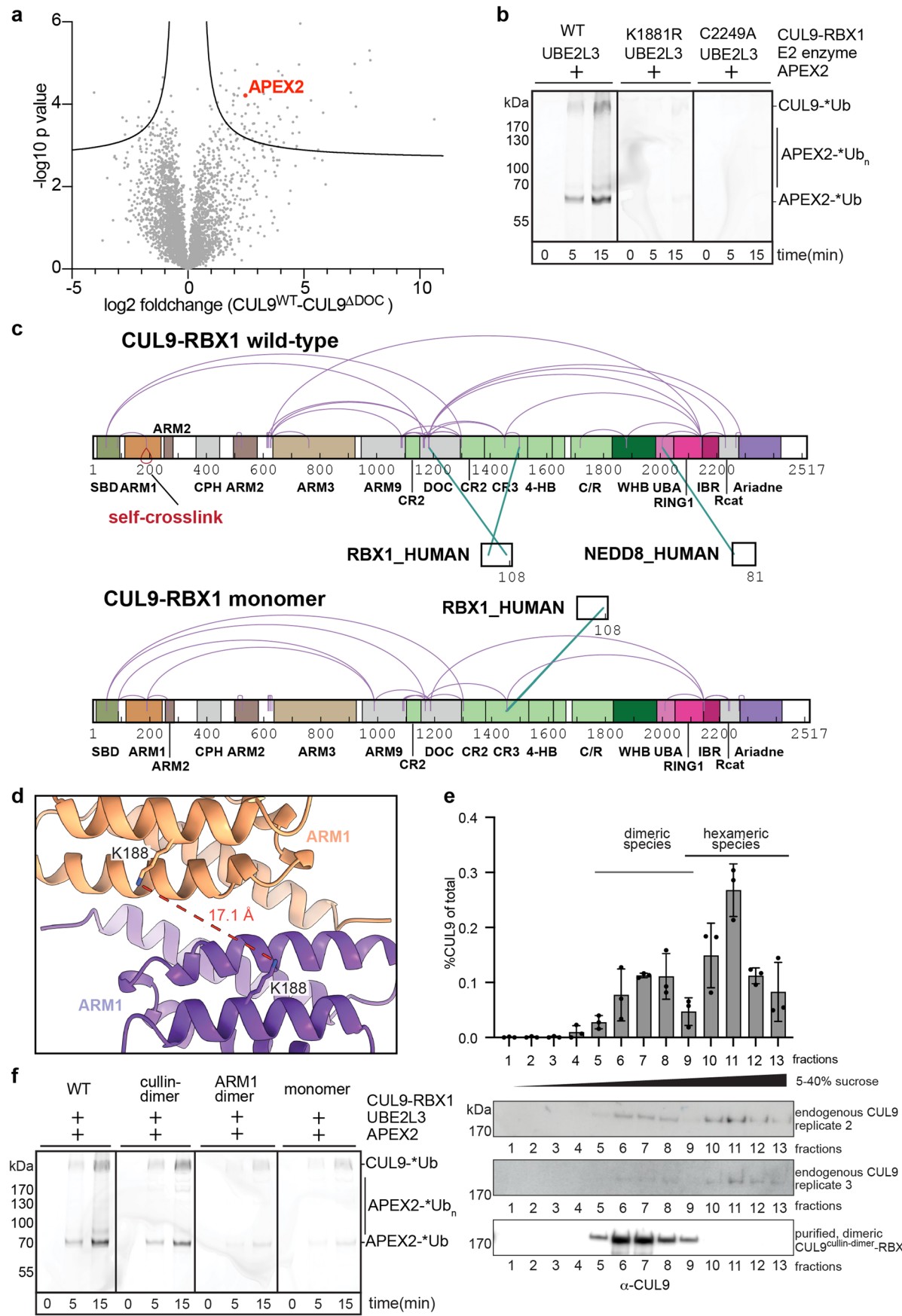

**Extended Data Fig. 8 | See next page for caption.**

**Extended Data Fig. 8 | Analysis of CUL9-RBX1 interactors and assembly and roles in ubiquitylation. a**, Comparison of proteins identified by quantitative mass spectrometry as interacting with TwinStrep-CUL9-RBX1 versus CUL9$^{\Delta DOC}$-RBX1 (variant in which DOC domain was replaced by a linker of sequence GSGSGSGS) affinity purified from HEK293S cells. Volcano plots of p-values (-log10) from two-tailed Student's t tests versus protein abundance (log2) differences. The significance curve was calculated based on a false-discovery-rate-adjusted P = 0.005 and a minimal fold change S0 = 0.1. Proteins above the curve significantly differ between WT CUL9-RBX1 and CUL9$^{\Delta DOC}$-RBX1. Data were obtained for each protein from three independent biological replicates. **b**, In vitro assays testing ubiquitylation of APEX2 by purified CUL9-RBX1 and CUL9-RBX1 variants with point mutations in the neddylation site (K1881) or ARIH-RBR catalytic cysteine (C2249). The assays detect fluorescently-labeled ubiquitin (Ub*) (*n* = 2 technically independent experiments). **c**, BS3 cross-linking mass spectrometry analysis of CUL9-RBX1 and the CUL9$^{monomer}$-RBX1 sample (a combination of the ARM1 dimer mutant which was made by replacing residues

1650–1690 with GSGSGSGS (ARM1 dimer) and the cullin-dimer mutant which was made by the two point mutations R125A Y152A (cullin-dimer)). 2D-Plots visualized with XiNET show crosslinks on schematic linear representations of the proteins in each complex. Table of crosslinks can be found in Supplementary table 5 and 6. **d**, Structural model of ARM1 dimerization interface, showing that crosslink between K188 from different CUL9 protomers - found only for WT CUL9-RBX1 - is consistent with hexameric assembly, but not a monomer. **e**, Immunoblot analysis of distribution of endogenous CUL9 protein in sucrose gradient fractions from U2OS cells, normalized relative to total CUL9. One replicate 1 is shown in Fig. 1f (n = 3 technically independent experiments, data are represented in columns as mean values +/− SD with individual data points indicated as dots). Sucrose gradient fractionation of the purified WT hexameric CUL9-RBX1 (Fig. 1f) and the cullin dimer mutant served as controls for migration. **f**, In vitro assays analyzing APEX2 ubiquitylation by indicated CUL9-RBX1 variants impaired for oligomerization (*n* = 2 technically independent experiments).

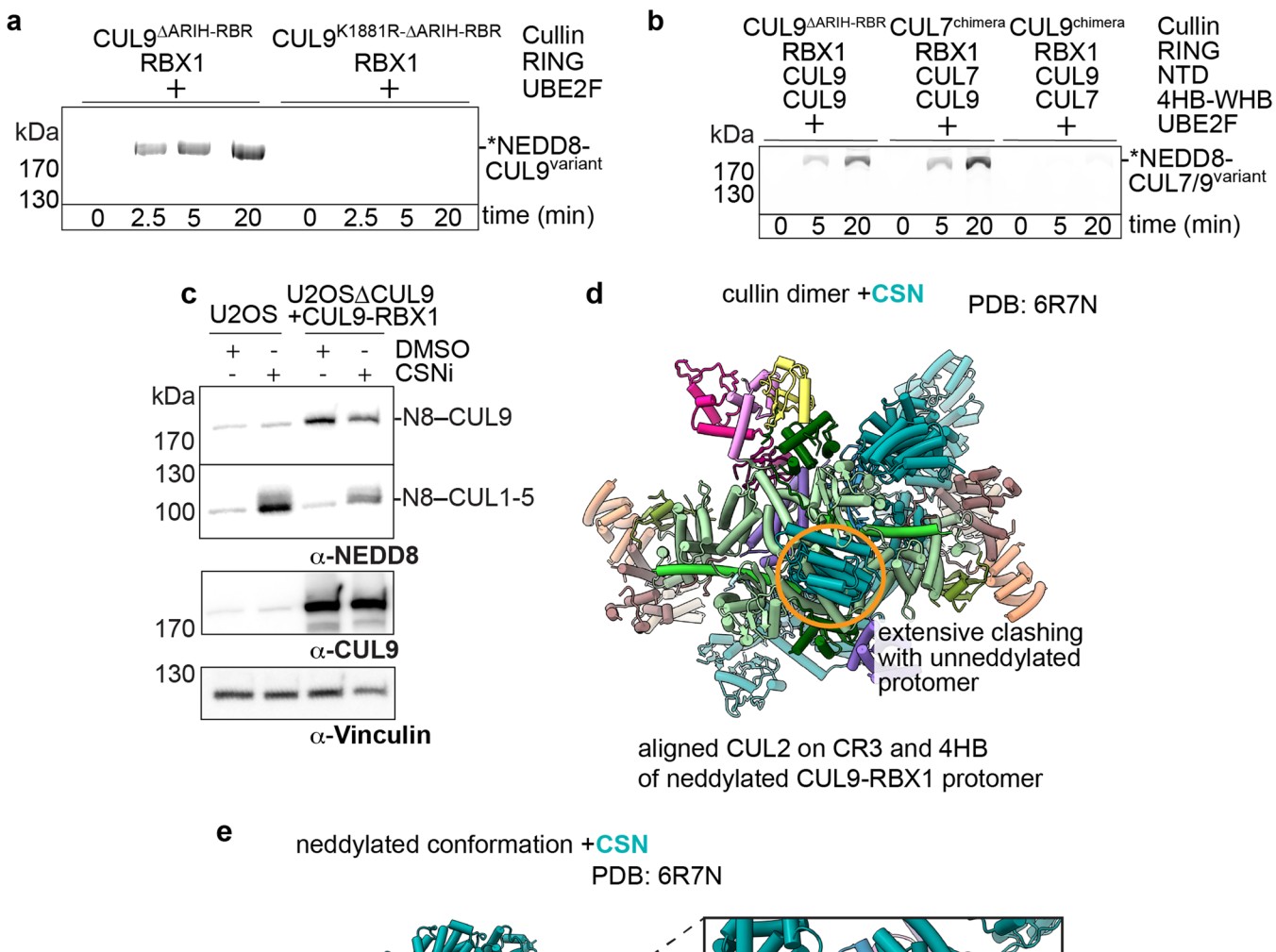

**Extended Data Fig. 9 | Structural and functional analysis of CUL9-RBX1 neddylation and deneddylation. a**, **b**, Purified neddylation machinery (NEDD8 E1 (NAE1-UBA3), NEDD8 E2 UBE2F) and the indicated CUL9-RBX1 complexes were used for in vitro neddylation assays, detecting fluorescently-labeled NEDD8 (*NEDD8) in SDS–PAGE gels (*n* = 2 technically independent experiments). **a**, CUL9-RBX1$^{\Delta ARIH-RBR}$ contains residues 1–1978 and lacks the ARIH-RBR element, CUL9-RBX1$^{K1881R-\Delta ARIH-RBR}$ is the same construct with the neddylation site K1881R substitution. **b**, CUL7-CUL9-RBX1 4HB-to-WHB chimera has the CUL7 sequence with the region spanning from the 4HB domain to the WHB domain swapped with the CUL9 sequence and CUL9-RBX1-CUL7 4HB-to-WHB chimera is the reciprocal swap of these domains. **c**, Treatment of parental U2OS cells, and CUL9 knock-out cells exogensously expressing CUL9 with CSN inhibitor (CSNi) or DMSO as control. Immunoblots detect NEDD8 (N8) linked to CUL9 or canonical cullins (CUL1-5), or total CUL9 or Vinculin as loading control (*n* = 2 technically independent experiments). **d**, **e**, Overlay of CSN-CUL2-RBX1 complex (PDB: 6R7N) and CUL9-RBX1 (this study) aligned on CR3 and 4HB of neddylated CUL9-RBX1 protomer. **d**, CSN modeled on neddylated protomer of mixed CUL9-RBX1 dimer is shown. **e**, CSN modeled only on neddylated protomer is shown, with closeup in inset on the right.

# Reporting Summary

## Statistics

For all statistical analyses, confirm that the following items are present in the figure legend, table legend, main text, or Methods section.

| n/a | Confirmed | |
|---|---|---|
| ☐ | ☒ | The exact sample size (*n*) for each experimental group/condition, given as a discrete number and unit of measurement |
| ☐ | ☒ | A statement on whether measurements were taken from distinct samples or whether the same sample was measured repeatedly |
| ☐ | ☒ | The statistical test(s) used AND whether they are one- or two-sided<br>*Only common tests should be described solely by name; describe more complex techniques in the Methods section.* |
| ☒ | ☐ | A description of all covariates tested |
| ☒ | ☐ | A description of any assumptions or corrections, such as tests of normality and adjustment for multiple comparisons |
| ☐ | ☒ | A full description of the statistical parameters including central tendency (e.g. means) or other basic estimates (e.g. regression coefficient) AND variation (e.g. standard deviation) or associated estimates of uncertainty (e.g. confidence intervals) |
| ☐ | ☒ | For null hypothesis testing, the test statistic (e.g. *F*, *t*, *r*) with confidence intervals, effect sizes, degrees of freedom and *P* value noted<br>*Give P values as exact values whenever suitable.* |
| ☒ | ☐ | For Bayesian analysis, information on the choice of priors and Markov chain Monte Carlo settings |
| ☒ | ☐ | For hierarchical and complex designs, identification of the appropriate level for tests and full reporting of outcomes |
| ☒ | ☐ | Estimates of effect sizes (e.g. Cohen's *d*, Pearson's *r*), indicating how they were calculated |

*Our web collection on statistics for biologists contains articles on many of the points above.*

## Software and code

Policy information about availability of computer code

| | |
|---|---|
| Data collection | Gel imaging: Amersham Imager 600, Amersham Typhoon;  Cryo-EM: SerialEM v3.8.0-b5, FEI EPU v2.7.0 |
| Data analysis | Assay Analysis: GraphPad Prism v9.2.0;  Cryo-EM: RELION v3.1, RELION 4.0, Gautomatch v0.56, CTFFIND v4.1;  Structure Analysis and Visualization: Chimera v1.13.1, ChimeraX v1.2.5; Model Building: COOT v0.8.9.1, Phenix.refine v1.17.1, DeepEMhancer version 2020.09.07 (https://github.com/rsanchezgarc/deepEMhancer); SEC-MALS: Wyatt Technology ASTRA v5.3; Massphotometry: Refeyn DiscoverMP v2.3.0; Mass spectrometry: Proteome Discoverer v2.5.0.400, cross-link analyzer v1.1.4, MaxQuant v2.2.0.0, MaxQuant v1.6.2.10, Python version v3.5.5 with packages numpy v1.21.5 and pandas v1.4.2. |

For manuscripts utilizing custom algorithms or software that are central to the research but not yet described in published literature, software must be made available to editors and reviewers. We strongly encourage code deposition in a community repository (e.g. GitHub). See the Nature Portfolio guidelines for submitting code & software for further information.

## Data

Policy information about availability of data

All manuscripts must include a data availability statement. This statement should provide the following information, where applicable:

- Accession codes, unique identifiers, or web links for publicly available datasets
- A description of any restrictions on data availability
- For clinical datasets or third party data, please ensure that the statement adheres to our policy

Cryo-EM maps will be available from the Electron Microscopy Data Bank, and the model coordinates will be available from Protein Data Bank upon publication: PDB ID 8Q7H (focused neddylated and unneddylated cullin dimer), PDB ID 8Q7E (hexameric assembly), PDB ID 8RHZ (unneddylated cullin dimer built in symmetry expanded map) and EMDB with codes EMD-18216 (focused neddylated and unneddylated cullin dimer), EMD-18214 (hexameric assembly), EMD-19179 (unneddylated cullin dimer symmetry expanded map), EMD-18218 (focused dimeric core), EMD-18217 (focused on E2-like density), EMD-18220 (CUL9ΔCPH-RBX1), EMD-18222 (CUL9ΔARM9-RBX1), EMD-18223 (CUL9ΔARIH-RBR-RBX1), EMD-18221 (CUL9ΔDOC-RBX1). The mass spectrometry data have been deposited to the ProteomeXchange Consortium (http://proteomecentral.proteomexchange.org) via the PRIDE repository with the dataset identifier PXD047326, PXD047229. Raw gels are provided as source data. Accession codes of published data which was used for comparison: PDB: 8Z7B, 7B5L, 1LDJ, 6V9I, 7ONI, 4P5O, 6R7N.

## Research involving human participants, their data, or biological material

Policy information about studies with human participants or human data. See also policy information about sex, gender (identity/presentation), and sexual orientation and race, ethnicity and racism.

| | |
|---|---|
| Reporting on sex and gender | N/A |
| Reporting on race, ethnicity, or other socially relevant groupings | N/A |
| Population characteristics | N/A |
| Recruitment | N/A |
| Ethics oversight | N/A |

Note that full information on the approval of the study protocol must also be provided in the manuscript.

# Field-specific reporting

Please select the one below that is the best fit for your research. If you are not sure, read the appropriate sections before making your selection.

☒ Life sciences    ☐ Behavioural & social sciences    ☐ Ecological, evolutionary & environmental sciences

For a reference copy of the document with all sections, see nature.com/documents/nr-reporting-summary-flat.pdf

# Life sciences study design

All studies must disclose on these points even when the disclosure is negative.

| | |
|---|---|
| Sample size | Sample size calculations were not performed. Selected sample sizes were designed to ensure clear and reliable interpretation of the results. Based on previous experience in terms of variability, at least two independent replicates were carried out for all functional assays. |
| Data exclusions | No data were excluded. |
| Replication | All experiments were performed at least twice, with numerous controls. All attempts at replication were successful. |
| Randomization | No grouped samples. |
| Blinding | No grouped samples. |

# Reporting for specific materials, systems and methods

We require information from authors about some types of materials, experimental systems and methods used in many studies. Here, indicate whether each material, system or method listed is relevant to your study. If you are not sure if a list item applies to your research, read the appropriate section before selecting a response.

## Materials & experimental systems

| n/a | Involved in the study |
|---|---|
| ☐ | ☒ Antibodies |
| ☐ | ☒ Eukaryotic cell lines |
| ☒ | ☐ Palaeontology and archaeology |
| ☒ | ☐ Animals and other organisms |
| ☒ | ☐ Clinical data |
| ☒ | ☐ Dual use research of concern |
| ☒ | ☐ Plants |

## Methods

| n/a | Involved in the study |
|---|---|
| ☒ | ☐ ChIP-seq |
| ☒ | ☐ Flow cytometry |
| ☒ | ☐ MRI-based neuroimaging |

## Antibodies

| | |
|---|---|
| Antibodies used | The antibodies against NEDD8 (#2745), UBE2M (#4913), β-Actin (#4967) were from Cell Signaling Technology. The antibodies against UBE2F (sc-398668) were from Santa Cruz Biotechnology. Anti-CUL9 antibody was a kind gift from Arno Alpi; this antibody was raised and validated by the MRC PPU Reagents and Services, School of Life Sciences, University of Dundee, Dundee, Scotland, DD1 5EH. The antibody against Vinculin (ab129002) was obtained from Abcam. |
| Validation | NEDD8 (https://www.cellsignal.com/products/primary-antibodies/nedd8-antibody/2745?_requestid=2891057)<br>UBE2M (https://www.cellsignal.com/products/primary-antibodies/ubc12-antibody/4913)<br>β-Actin (https://www.cellsignal.com/products/primary-antibodies/b-actin-antibody/4967)<br>UBE2F (https://www.scbt.com/de/p/ube2f-antibody-c-11)<br>CUL9 (https://mrcppureagents.dundee.ac.uk/ and this study)<br>Vinculin (https://www.abcam.com/products/primary-antibodies/vinculin-antibody-epr8185-ab129002.html) |

## Eukaryotic cell lines

Policy information about cell lines and Sex and Gender in Research

| | |
|---|---|
| Cell line source(s) | HEK293S GnTI- (identifier: CRL-3022), U2OS (identifier: HTB-96) were obtained from ATCC. U2OS CUL9 knockout cell line was a kind gift by Yue Xiong. Sf9 cells were obtained from Thermo Fischer (identifier: 11496015). |
| Authentication | Cell lines were not authenticated. |
| Mycoplasma contamination | Cell lines were periodically tested for Mycoplasma contamination and were always negative. |
| Commonly misidentified lines<br>(See ICLAC register) | No commonly misidentified cell lines were used in this study. |

