## [Peer Review File · Nature Structural & Molecular Biology]

Peer Review Information

Manuscript Title: Noncanonical assembly, neddylation, and chimeric cullin-RING/RBR ubiquitylation by the 1.8 MDa CUL9 E3 ligase complex

Corresponding author name(s): Brenda Schulman

Reviewer Comments & Decisions:

Decision Letter, initial version:
--

Message: 5th Oct 2023

Dear Professor Schulman,

Thank you again for submitting your manuscript "Noncanonical assembly, neddylation, and chimeric cullin-RING/RBR ubiquitylation by the 1.8 MDa CUL9 E3 ligase complex". I apologise for the delay in responding, which resulted from the difficulty in obtaining suitable referee reports. Nevertheless, we now have comments (below) from the 3 reviewers who evaluated your paper. In light of these reports, we remain interested in your study and would like to see your response to the comments of the referees, in the form of a revised manuscript.

You will see that all referees appreciate the structural and biochemical data, finding that they adequately support most of the proposed mechanistic conclusions. They additionally appreciate the insight and progress imparted in this work. However, there are some points that require addressing in a revised manuscript. More specifically, both Reviewer #1 and #3 question the lack of information with respect to the culprit E2 and seem to think that obtaining such information would further boost the manuscript. The same reviewers pose a few mechanistic/functional questions (unassigned density at the center of the complex, cellular distribution of hexamer versus dimer, Cul7 affecting Cul9 (de)neddylation, potential disease-associated Cul9 mutants that map to the oligomerization or other relevant from this study domains), which, we editorially agree, that would further elevate the value of this work. Finally, the referees bring up a few points that require clarifications and textual changes.

Please be sure to address/respond to all concerns of the referees in full in a point-by-point response and highlight all changes in the revised manuscript text file. If you have comments that are intended for editors only, please include those in a separate cover letter.

We expect to see your revised manuscript within 3 months. If you cannot send it within this time, please contact us to discuss an extension; we would still consider your revision, provided that no similar work has been accepted for publication at NSMB or published elsewhere.

Reporting Summary:

SOURCE DATA: we urge authors to provide, in tabular form, the data underlying the graphical representations used in figures. This is to further increase transparency in data reporting, as detailed in this editorial (<http://www.nature.com/nsmb/journal/v22/n10/full/nsmb.3110.html>). Spreadsheets can be submitted in excel format. Only one (1) file per figure is permitted; thus, for multi-paneled figures, the source data for each panel should be clearly labeled in the Excel file; alternately the data can be provided as multiple, clearly labeled sheets in an Excel file. When submitting files, the title field should indicate which figure the source data pertains

to. We encourage our authors to provide source data at the revision stage, so that they are part of the peer-review process.

Data availability: this journal strongly supports public availability of data. All data used in accepted papers should be available via a public data repository, or alternatively, as Supplementary Information. If data can only be shared on request, please explain why in your Data Availability Statement, and also in the correspondence with your editor. Please note that for some data types, deposition in a public repository is mandatory - more information on our data deposition policies and available repositories can be found below: <https://www.nature.com/nature-research/editorial-policies/reporting-standards#availability-of-data>

[Redacted]

Sincerely,

Dimitris Typas

Associate Editor
Nature Structural & Molecular Biology
ORCID: 0000-0002-8737-1319

Referee expertise:

Referee #1: structural ubiquitylation

Referee #2: UPS, chemical biology ubiquitylation

Referee #3: structural ubiquitylation

Reviewers' Comments:

Reviewer #1:

Remarks to the Author:

In this study, Horn-Ghetko et al. determined and presented the cryo-EM structure of CUL9 complex, which is an atypical Cullin protein with RBR domains and forms an unexpected hexamer. The team further dissected the oligomerization interface, reconstituted ubiquitination activity with native substrate, TP53, and investigated the neddylation / deneddylation regulation of CUL9.

In summary, this is an exceptionally well-executed work with nicely presented structural and biochemical data. It joins the other CRL work previously published by the same group, providing critical insights into the structure, mechanism, and regulation of CUL9, while also contributing to our broader comprehension of large E3 ligase complexes. This work certainly serves as a foundation for further investigations.

While there are no major concerns, outlined below is a list of minor points that the authors may consider clarifying to enhance the manuscript:

1. The description of individual domains of CUL9 in the first section of Results appeared before the introduction of Fig. 2a. This could be difficult for readers who are not familiar with CUL9.
2. Does the subpeak in mass photometry (Fig. 2e) suggest a dimer? From the biochemical assays (Fig. 4c), it looks like one potential dimer form is actually active. Is this biologically relevant? In other words, what is the distribution of hexamer vs. dimer in a cell? From the cryo-EM processing flowchart, it looks like the dimer form is negligible.
3. Have the authors tried symmetry expansion which in principle could provide 3x more particle for focused classification and refinement?
4. Line 148: a broken reference.
5. It is surprising that E2 could not be identified by mass spectrometry from the purified complex. Was it because the peptide fragments do not support identification or no E2 was detected at all?

6. The unassigned density at the center of the complex (Fig. 3b) is puzzling. Is it due to applying the symmetry? Were any candidates identified by MS? DOC deletion of CUL9 does not affect ubiquitination activity (Ex Fig. 7c), and it still forms a hexamer, which suggested an unknown function of the central part.

7. Typo in the author name: "Yue Xiong", not "Xue Yiong".

Reviewer #2:

Remarks to the Author:

In this manuscript, Horn-Ghetko et al solve the hexameric structure of CUL9 E3-ligase complex, and perform detailed biochemical investigation into its interactions with E3-activity regulating enzymes (UBE2M, UBE2F, CSN), and substrates (TP53). Further, they used mutagenesis and domain deletion to investigate the functional roles of each domain in controlling complex assembly and activation. In doing so, they discover not only a novel structure, but also unveil new E3-ligase assembly and regulation mechanisms. The biochemistry is performed to a high standard and appropriate controls and orthogonal methods used to validate findings. The text is clearly written, although speculative discussion is included, it is clearly and appropriately differentiated from the manuscript claims and serves to put the data in context of what is currently known in the field.

Reviewer #3:

Remarks to the Author:

This study centers around a remarkable structure of the chimeric cullin-RING/RBR E3 ligase CUL9. The structure reveals a complex oligomerization that is required for full ligase activity. Due to its purification from HEK cells, the CUL9 structure also provides visualization of both the neddylated and unneddylated forms, as well as co-purification of an E2 ubiquitin-conjugating enzyme bound to the RBR RING1 domain. In addition to these molecular insights into CUL9 function, the authors also present important discoveries into the regulation of CUL9 neddylation. Overall, the manuscript is well written, the figures are clear, and the experiments are carefully controlled. After consideration of the comments below, this work is well suited for publication in NSMB.

- 1) Could the authors address/speculate why no crosslinks were observed between UBE2L3 and the RING1 domain?
- 2) Is it possible that CUL7 binding affects accessibility to CUL9 neddylation/deneddylation?
- 3) Are there any disease-associated mutations in CUL9 that map to the oligomerization interfaces?

Minor comments:

- 1) There is a typo in the figure legend of Extended Data Fig. 1f.
- 2) A reference is not formatted correctly on Line 148.
- 3) It might be helpful to label the RTI Helix in Figures 2g-h.
- 4) Line 223 has a duplicated "with".
- 5) Line 258, "these" should be "the".
- 6) Lines 265 and 270, check that the figure references are correct.
- 7) Line 324, an "and" is missing after CSN.

- 8) Line 324, check that the figure reference is correct.
- 9) Legend of Figure 6 mentions DEN1, whereas SENP8 nomenclature is used elsewhere.

Author Rebuttal to Initial comments

Responses to Reviewer comments are in blue.

Reviewer #1:

Remarks to the Author:

In this study, Horn-Ghetko et al. determined and presented the cryo-EM structure of CUL9 complex, which is an atypical Cullin protein with RBR domains and forms an unexpected hexamer. The team further dissected the oligomerization interface, reconstituted ubiquitination activity with native substrate, TP53, and investigated the neddylation / deneddylation regulation of CUL9.

In summary, this is an exceptionally well-executed work with nicely presented structural and biochemical data. It joins the other CRL work previously published by the same group, providing critical insights into the structure, mechanism, and regulation of CUL9, while also contributing to our broader comprehension of large E3 ligase complexes. This work certainly serves as a foundation for further investigations.

We thank the reviewer for such kind comments and enthusiasm for our study!

While there are no major concerns, outlined below is a list of minor points that the authors may consider clarifying to enhance the manuscript:

1. The description of individual domains of CUL9 in the first section of Results appeared before the introduction of Fig. 2a. This could be difficult for readers who are not familiar with CUL9.

We moved the description of the domains to the part of the text where we introduce Fig. 2a, and have slightly edited this as follows:

'The visible regions from CUL9 include the small beta domain (SBD), the ARM element (ARM1-ARM3 domains), the cullin element (CR2, CR3, 4HB, C/R, and WHB domains) and ARIH1-RBR element. RBX1 has two domains: an N-terminal strand embedded in CUL9's C/R domain is tethered to the C-terminal RING domain. Fitting the dimeric subcomplex into the full map showed details of the hexameric assembly (Fig. 1c and 2a-d).'

2. Does the subpeak in mass photometry (Fig. 2e) suggest a dimer? From the biochemical assays (Fig. 4c), it looks like one potential dimer form is actually active. Is this biologically relevant? In other words, what is the distribution of hexamer vs. dimer in a cell? From the cryo-EM processing flowchart, it looks like the dimer form is negligible.

We have addressed this in two ways. First, we note in the text the small fraction of the purified complex that corresponds to a dimer in mass photometry, as follows.

'Notably, endogenous CUL9 was also detected in preceding fractions, indicating the presence of lower order oligomers such as dimers, as similarly observed in mass photometry (Fig. 1e-f, Extended Data Fig. 1d).'

Second, we examined the distribution of dimer versus hexamer in cells. Briefly, we repeated sucrose gradient fractionation from cells so as to have three biological replicates for quantification. We quantified the immunoblot signal across the fractions. We compare these results to sucrose gradient fractionation of the purified WT hexamer and the active dimer mutant as controls for migration. These new data are shown in Figures 1f and Extended Data Figure 8e in the revised manuscript, and below.

Figure 1e-f:

New next:

'Purification of dimeric complexes allowed re-evaluation of oligomerization status of endogenous CUL9. Comparing migration in sucrose gradients suggests that some cellular CUL9 is hexameric, while a smaller fraction aligns with a dimer (Fig. 1f, Extended Data Fig. 8e).'

Extended Data Fig. 8e:

3. Have the authors tried symmetry expansion which in principle could provide 3x more particle for focused classification and refinement?

Indeed, based on this suggestion, symmetry expansion enabled our obtaining a structure for the inactive cullin dimer. Symmetry expansion of 301279 particles after C3 homogenous refinement to 903837 particles, followed by 3D classifications to exclude particles with the active conformation and local refinements on one dimeric subunit yielded a 3.37 Å resolution map of a cullin dimer. Here, both protomers are unneddylated.

We refer to this in the revised text as follows:

'Superimposing homologous regions of Protomer A (unneddylated) on Protomer B (neddylated) and vice-versa show that the hexamer could be formed by either a fully unneddylated or neddylated complex, the former also observed in a map obtained through symmetry expansion (Extended Data Fig. 3c-d).'

Figure 3c: **c** unneddylated cullin dimer symmetry expanded map

Unfortunately, despite several different attempts, we were unable to obtain better maps for neddylated CUL9-RBX1, or showing E2 binding, using symmetry expansion.

4. Line 148: a broken reference.

Thanks for spotting this! We fixed this in the revision.

5. It is surprising that E2 could not be identified by mass spectrometry from the purified complex. Was it because the peptide fragments do not support identification or no E2 was detected at all?

To address this and requests from the editors, we performed Twin Strep Affinity Purification-Mass Spectrometry (AP-MS) experiments comparing proteins co-purifying with WT CUL9-RBX1 versus the CUL9^{ΔRING1}-RBX1 mutant lacking the E2-binding RING1 domain. Of 11 ubiquitin conjugating E2 enzymes identified, the only E2 whose association depended on CUL9's RING1 domain was UBE2L3. The new data are shown in a volcano plot in Extended Data Fig. 5b of the revised manuscript, and have been deposited to the PRIDE database with accession code PXD047229 (Username: reviewer_pxd047229@ebi.ac.uk Password: hdQQn3yG). We refer to the new data as follows:

'We modeled the E2 as UBE2L3 based on: (1) our isothermal titration calorimetry showing UBE2L3, but not a UBE2D-family E2, binding the CUL9 RBR element; (2) affinity purification-mass spectrometry (AP-MS) data showing RING1-dependent endogenous UBE2L3 association with CUL9 ectopically-expressed in HEK293S cells; (3) prior data showing CUL9 binds UBE2L3^{11,50,51}; and (4) a predilection for RBR E3s to employ this E2⁴⁹ (Extended Data Fig. 5a-b).'

6. The unassigned density at the center of the complex (Fig. 3b) is puzzling. Is it due to applying the symmetry?

The central density is visible in maps generated without applying symmetry.

Were any candidates identified by MS? DOC deletion of CUL9 does not affect ubiquitination activity (Ex Fig. 7c), and it still forms a hexamer, which suggested an unknown function of the central part.

When we launched experiments to address this, we did not have any expectations for potential outcomes. The results were an exciting surprise, and we hope the reviewer finds them as interesting as we do.

Briefly, we performed AP-MS comparing interaction partners of WT CUL9-RBX1 with the mutant lacking the DOC domain (the same mutant that lacks the central density in Figure 3g). Cross-referencing our list of hits with CUL9 interactors reported in the BioGRID database (<https://thebiogrid.org>) showed a single candidate on both: APEX2. APEX2 ranks #4 in BioGRID in frequency of identification as a CUL9 interactor. Since the top 3-ranked listings in BioGRID are bona fide CUL9 interactors (TP53, CUL7 and RBX1), we pursued APEX2 as a candidate. Indeed, APEX2 that copurified with co-expressed CUL9-RBX1 served as substrate for in vitro ubiquitylation (Figure 3i).

We next compared the domain dependencies for in vitro ubiquitylation of purified APEX2 versus those for TP53. APEX2 ubiquitylation depends on CUL9's DOC domain while TP53 ubiquitylation does not. Furthermore, unlike TP53, APEX2 ubiquitylation does not depend on the CPH domain (Figure 3i).

In addition to these figures (shown below), we uploaded the AP-MS data to the PRIDE database with accession code PXD047229 (Username: reviewer_pxd047229@ebi.ac.uk Password: hdQQn3yG), and have added the following to the Results section:

'To gain insights into a potential role for the DOC domain, we compared interactors of CUL9-RBX1 versus CUL9^{ΔDOC}-RBX1. Cross-referencing our AP-MS hits (Extended Data Fig. 8a) with CUL9 interactors reported by BioGRID⁵³ revealed a single top hit in both: APEX2. Indeed, APEX2 was ubiquitylated in vitro by neddylated CUL9-RBX1, depending on CUL9's DOC domain, neddylation site (K1881) and RBR catalytic cysteine (C2249) (Fig. 3i, Extended Data Fig. 8b). Notably, APEX2 ubiquitylation was unaffected by deletion of CUL9's CPH domain, while TP53 was subject to ubiquitylation by the CUL9^{ΔDOC}-RBX1 mutant. Although future studies will be required to determine the biological functions of APEX2 ubiquitylation by CUL9-RBX1, we note that its enzymatic activity as an apurinic/aprimidinic endodeoxyribonuclease is in line with previous findings that CUL9 plays roles in maintaining genome integrity^{13,15,17}.'

Extended Data Fig. 8a-b:

Fig. 3i:

Also, we show that oligomerization is required in Extended Data Fig. 8f:

7. Typo in the author name: “Yue Xiong”, not “Xue Yiong”.

Oh, that is a major typo! Thank you! We made the correction.

Reviewer #2:

Remarks to the Author:

In this manuscript, Horn-Ghetko et al solve the hexameric structure of CUL9 E3-ligase complex, and perform detailed biochemical investigation into its interactions with E3-activity regulating enzymes (UBE2M, UBE2F, CSN), and substrates (TP53). Further, they used mutagenesis and domain deletion to investigate the functional roles of each domain in controlling complex assembly and activation. In doing so, they discover not only a novel structure, but also unveil new E3-ligase assembly and regulation mechanisms. The biochemistry is performed to a high standard and appropriate controls and orthogonal methods used to validate findings. The text is clearly written, although speculative discussion is included, it is clearly and appropriately differentiated from the manuscript claims and serves to put the data in context of what is currently known in the field.

We thank the reviewer for such kind comments and enthusiasm for our study!

Reviewer #3:

Remarks to the Author:

This study centers around a remarkable structure of the chimeric cullin-RING/RBR E3 ligase CUL9. The structure reveals a complex oligomerization that is required for full ligase activity. Due to its purification from HEK cells, the CUL9 structure also provides visualization of both the neddylated and unneddylated forms, as well as co-purification of an E2 ubiquitin-conjugating enzyme bound to the RBR RING1 domain. In addition to these molecular insights into CUL9 function, the authors also present important discoveries into the regulation of CUL9 neddylation. Overall, the manuscript is well written, the figures are clear, and the experiments are carefully controlled. After consideration of the comments below, this work is well suited for publication in NSMB.

We thank the reviewer for such kind comments and enthusiasm for our study!

1) Could the authors address/speculate why no crosslinks were observed between UBE2L3 and the RING1 domain?

The sole lysine (K2094) in the RING1 domain of CUL9 points away from the E2, and is spatially obstructed by the E2-binding portions of the RING1 domain. Nonetheless, the E2-linked ubiquitin does form crosslinks with the E2 and the RING1-to-IBR (RTI) helix, in agreement with the structural model for an E2~ubiquitin-CUL9 complex shown here.

2) Is it possible that CUL7 binding affects accessibility to CUL9 neddylation/denedylation?

To address this, we tested if adding CUL7-RBX1 or CUL7-RBX1-FBXW8-SKP1 (i.e. CRL7^{FBXW8}) affects CUL9-RBX1 neddylation and saw no effect. We also tried co-expressing CUL7-RBX1 with CUL9-RBX1, but this also did not alter neddylation. The results are shown below for the reviewer.

3) Are there any disease-associated mutations in CUL9 that map to the oligomerization interfaces?

The cBio Cancer Genomics Portal lists 365 CUL9 missense mutations in various cancers. Notably, CUL9 mutations were found in more than 10% of cases of Stomach Adenocarcinoma, Skin Cutaneous Melanoma and Uterine Corpus Endometrial Carcinoma.

https://www.cbioportal.org/results/cancerTypesSummary?case_set_id=all&gene_list=CUL9&cancer_study_list=5c8a7d55e4b046111fee2296&comparison_selectedGroups=%5B%22Altered%20group%22%2C%22Unaltered%20group%22%5D&comparison_subtab=survival.

However, when we mapped these on the CUL9 structure, there was no clear pattern. They do not cluster at dimerization interfaces nor to any other region. Instead the mutations are distributed over the CUL9 structure as shown in the figure for reviewers.

Cancer mutants mapped on CUL9

Minor comments:

- 1) There is a typo in the figure legend of Extended Data Fig. 1f.
- 2) A reference is not formatted correctly on Line 148.
- 3) It might be helpful to label the RTI Helix in Figures 2g-h.
- 4) Line 223 has a duplicated "with".
- 5) Line 258, "these" should be "the".
- 6) Lines 265 and 270, check that the figure references are correct.
- 7) Line 324, an "and" is missing after CSN.
- 8) Line 324, check that the figure reference is correct.
- 9) Legend of Figure 6 mentions DEN1, whereas SENP8 nomenclature is used elsewhere.

We fixed all these issues in the revised manuscript.

Decision Letter, first revision:

Message: Our ref: NSMB-A48160A

4th Jan 2024

Dear Professor Schulman,

Thank you for submitting your revised manuscript "Noncanonical assembly, neddylation, and chimeric cullin-RING/RBR ubiquitylation by the 1.8 MDa CUL9 E3 ligase complex" (NSMB-A48160A). It has now been seen by the original referees and their comments are below. The reviewers find that the paper has further improved in revision, and therefore we are happy to accept it in principle in Nature Structural & Molecular Biology, pending minor revisions to satisfy the referees' final requests and to comply with our editorial and formatting guidelines.

We are now performing detailed checks on your paper and will send you a checklist detailing our editorial and formatting requirements in about two weeks. Please do not upload the final materials and make any revisions until you receive this additional information from us.

To facilitate our work at this stage, it is important that we have a copy of the main text as a word file. If you could please send along a word version of this file as soon as possible, we would greatly appreciate it; please make sure to copy the NSMB account (cc'ed above).

Sincerely,

Dimitris Typas
Associate Editor
Nature Structural & Molecular Biology
ORCID: 0000-0002-8737-1319

Reviewer #1 (Remarks to the Author):

In the revised manuscript, the authors have effectively addressed all of my concerns, presenting beautifully designed additional experiments. I truly appreciate the considerable efforts the authors have invested in enhancing the manuscript, and I wholeheartedly endorse the publication of this paper.

Reviewer #2 (Remarks to the Author):

The authors have address concerns and I support publication in NSMB

Reviewer #3 (Remarks to the Author):

The authors have nicely addressed my concerns and the manuscript is significantly improved. My only remaining suggestion would be to alter the section heading "Oligomeric assembly contributes to TP53 ubiquitylation" to instead more broadly refer to substrate ubiquitylation, as the authors now also present data on APEX2.

Author Rebuttal, first revision:

NSMB-A48160A

Noncanonical assembly, neddylation, and chimeric cullin-RING/RBR ubiquitylation by the 1.8 MDa CUL9 E3 ligase complex

Responses to Reviewers in blue.

Reviewer #1 (Remarks to the Author):

In the revised manuscript, the authors have effectively addressed all of my concerns, presenting beautifully designed additional experiments. I truly appreciate the considerable efforts the authors have invested in enhancing the manuscript, and I wholeheartedly endorse the publication of this paper.

We really appreciate your thoughtful and enthusiastic response!

Reviewer #2 (Remarks to the Author):

The authors have address concerns and I support publication in NSMB

We are very pleased by your enthusiasm for our study!

Reviewer #3 (Remarks to the Author):

The authors have nicely addressed my concerns and the manuscript is significantly improved. My only remaining suggestion would be to alter the section heading "Oligomeric assembly contributes to TP53 ubiquitylation" to instead more broadly refer to substrate ubiquitylation, as the authors now also present data on APEX2.

We are very pleased by your enthusiasm for our study! We changed the section heading according to this helpful suggestion.

Final Decision Letter:

Message: 26th Feb 2024

Dear Professor Schulman,

We are now happy to accept your revised paper "Noncanonical assembly, neddylation, and chimeric cullin-RING/RBR ubiquitylation by the 1.8 MDa CUL9 E3 ligase complex" for publication as an Article in Nature Structural & Molecular Biology.

Your paper will be published online soon after we receive proof corrections and will appear in print in the next available issue. You can find out your date of online publication by contacting the production team shortly after sending your proof corrections.

You may wish to make your media relations office aware of your accepted publication, in

case they consider it appropriate to organize some internal or external publicity. Once your paper has been scheduled you will receive an email confirming the publication details. This is normally 3-4 working days in advance of publication. If you need additional notice of the date and time of publication, please let the production team know when you receive the proof of your article to ensure there is sufficient time to coordinate. Further information on our embargo policies can be found here: <https://www.nature.com/authors/policies/embargo.html>

Please note that *Nature Structural & Molecular Biology* is a Transformative Journal (TJ). Authors may publish their research with us through the traditional subscription access route or make their paper immediately open access through payment of an article-processing charge (APC). Authors will not be required to make a final decision about access to their article until it has been accepted. Find out more about Transformative Journals

You will not receive your proofs until the publishing agreement has been received through

our system.

Sincerely,

Dimitris Typas
Associate Editor
Nature Structural & Molecular Biology
ORCID: 0000-0002-8737-1319